# Convergence of Meta-Learning with Task-Specific Adaptation over Partial Parameters

**Kaiyi Ji**
Department of ECE
The Ohio State University
ji.367@osu.edu

**Jason D. Lee**
Department of EE
Princeton University
jasonlee@princeton.edu

**Yingbin Liang**
Department of ECE
The Ohio State University
liang.889@osu.edu

**H. Vincent Poor**
Department of EE
Princeton University
poor@princeton.edu

## Abstract

Although model-agnostic meta-learning (MAML) is a very successful algorithm in meta-learning practice, it can have high computational cost because it updates all model parameters over both the inner loop of task-specific adaptation and the outer-loop of meta initialization training. A more efficient algorithm ANIL (which refers to almost no inner loop) was proposed recently by Raghu et al. 2019, which adapts only a small subset of parameters in the inner loop and thus has substantially less computational cost than MAML as demonstrated by extensive experiments. However, the theoretical convergence of ANIL has not been studied yet. In this paper, we characterize the convergence rate and the computational complexity for ANIL under two representative inner-loop loss geometries, i.e., strongly-convexity and nonconvexity. Our results show that such a geometric property can significantly affect the overall convergence performance of ANIL. For example, ANIL achieves a faster convergence rate for a strongly-convex inner-loop loss as the number $N$ of inner-loop gradient descent steps increases, but a slower convergence rate for a nonconvex inner-loop loss as $N$ increases. Moreover, our complexity analysis provides a theoretical quantification on the improved efficiency of ANIL over MAML. The experiments on standard few-shot meta-learning benchmarks validate our theoretical findings.

## 1 Introduction

As a powerful learning paradigm, meta-learning [2, 29] has recently received significant attention, especially with the incorporation of training deep neural networks [5, 30]. Differently from the conventional learning approaches, meta-learning aims to effectively leverage the datasets and prior knowledge of a task ensemble in order to rapidly learn new tasks often with a small amount of data such as in few-shot learning. A broad collection of meta-learning algorithms have been developed so far, which range from metric-based [16, 28], model-based [21, 30], to optimization-based algorithms [5, 22]. The focus of this paper is on the optimization-based approach, which is often easy to be integrated with optimization formulations of many machine learning problems.

One highly successful optimization-based meta-learning approach is the model-agnostic meta-learning (MAML) algorithm [5], which has been applied to many application domains including classification [25], reinforcement learning [5], imitation learning [9], etc. At a high level, the MAML algorithm consists of two optimization stages: the inner loop of task-specific adaptation and the outer

(meta) loop of initialization training. Since the outer loop often adopts a gradient-based algorithm, which takes the gradient over the inner-loop algorithm (i.e., the inner-loop optimization path), even the simple inner loop of gradient descent updating can result in the Hessian update in the outer loop, which causes significant computational and memory cost. Particularly in deep learning, if all neural network parameters are updated in the inner loop, then the cost for the outer loop is extremely high. Thus, designing simplified MAML, especially the inner loop, is highly motivated. ANIL (which stands for *almost no inner loop*) proposed in [24] has recently arisen as such an appealing approach. In particular, [24] proposed to update only a small subset (often only the last layer) of parameters in the inner loop. Extensive experiments in [24] demonstrate that ANIL achieves a significant speedup over MAML without sacrificing the performance.

Despite extensive empirical results, there has been no theoretical study of ANIL yet, which motivates this work. In particular, we would like to answer several new questions arising in ANIL (but not in the original MAML). While the outer-loop loss function of ANIL is still nonconvex as MAML, the inner-loop loss can be either *strongly convex* or *nonconvex* in practice. The strong convexity occurs naturally if only the last layer of neural networks is updated in the inner loop, whereas the nonconvexity often occurs if more than one layer of neural networks are updated in the inner loop. Thus, our theory will explore how such different geometries affect the convergence rate, computational complexity, as well as the hyper-parameter selections. We will also theoretically quantify how much computational advantage ANIL achieves over MAML by training only partial parameters in the inner loop.

## 1.1 Summary of Contributions

In this paper, we characterize the convergence rate and the computational complexity for ANIL with $N$-step inner-loop gradient descent, under nonconvex outer-loop loss geometry, and under two representative inner-loop loss geometries, i.e., strongly-convexity and nonconvexity. Our analysis also provides theoretical guidelines for choosing the hyper-parameters such as the stepsize and the number $N$ of inner-loop steps under each geometry. We summarize our specific results as follows.

- **Convergence rate**: ANIL converges sublinearly with the convergence error decaying sublinearly with the number of sampled tasks due to nonconvexity of the meta objective function. The convergence rate is further significantly affected by the geometry of the inner loop. Specifically, ANIL converges exponentially fast with $N$ initially and then saturates under the strongly-convex inner loop, and constantly converges slower as $N$ increases under the nonconvex inner loop.

- **Computational complexity**: ANIL attains an $\epsilon$-accurate stationary point with the gradient and second-order evaluations at the order of $\mathcal{O}(\epsilon^{-2})$ due to nonconvexity of the meta objective function. The computational cost is also significantly affected by the geometry of the inner loop. Specifically, under the strongly-convex inner loop, its complexity first decreases and then increases with $N$, which suggests a moderate value of $N$ and a constant stepsize in practice for a fast training. But under the nonconvex inner loop, ANIL has higher computational cost as $N$ increases, which suggests a small $N$ and a stepsize at the level of $1/N$ for desirable training.

- Our experiments validate that ANIL exhibits aforementioned *very different* convergence behaviors under the two inner-loop geometries.

From the technical standpoint, we develop new techniques to capture the properties for ANIL, which does not follow from the existing theory for MAML [4, 14]. First, our analysis explores how different geometries of the inner-loop loss (i.e., strongly-convexity and nonconvexity) affect the convergence of ANIL. Such comparison does not exist in MAML. Second, ANIL contains parameters that are updated only in the outer loop, which exhibit *special* meta-gradient properties not captured in MAML.

## 1.2 Related Works

**MAML-type meta-learning approaches.** As a pioneering meta-initialization approach, MAML [5] aims to find a good initialization point such that a few gradient descent steps starting from this point achieves fast adaptation. MAML has inspired various variant algorithms [6, 7, 8, 12, 20, 24, 25, 33]. For example, FOMAML [5] and Reptile [22] are two first-order MAML-type algorithms which avoid second-order derivatives. [7] provided an extension of MAML to the online setting. Based on the implicit differentiation technique, [25] proposed a MAML variant named iMAML by formulating the inner loop as a regularized empirical risk minimization problem. More recently, [24] modifies

MAML to ANIL by adapting a small subset of model parameters during the inner loop in order to reduce the computational and memory cost. This paper provides the theoretical guarantee for ANIL as a complement to its empirical study in [24].

**Other optimization-based meta-learning approaches.** Apart from MAML-type meta-initialization algorithms, another well-established framework in few-shot meta learning [3, 18, 26, 28, 32] aims to learn good parameters as a common embedding model for all tasks. Building on the embedded features, task-specific parameters are then searched as a minimizer of the inner-loop loss function [3, 18]. Compared to ANIL, such a framework does not train the task-specific parameters as initialization, whereas ANIL trains a good initialization for the task-specific parameters.

**Theory for MAML-type approaches.** There have been only a few studies on the statistical and convergence performance of MAML-type algorithms. [6] proved a universal approximation property of MAML under mild conditions. [25] analyzed the convergence of iMAML algorithm based on implicit meta gradients. [4] analyzed the convergence of one-step MAML for a nonconvex objective, and [14] analyzed the convergence of multi-step MAML in the nonconvex setting. As a comparison, we analyze the ANIL algorithm provided in [24], which has different properties from MAML due to adapting only partial parameters in the inner loop.

**Notations.** For a function $L(w, \phi)$ and a realization $(w', \phi')$, we define $\nabla_w L(w', \phi') = \frac{\partial L(w, \phi)}{\partial w}\big|_{(w', \phi')}$, $\nabla_w^2 L(w', \phi') = \frac{\partial^2 L(w, \phi)}{\partial w^2}\big|_{(w', \phi')}, \nabla_\phi \nabla_w L(w', \phi') = \frac{\partial^2 L(w, \phi)}{\partial \phi \partial w}\big|_{(w', \phi')}$. The same notations hold for $\phi$.

## 2 Problem Formulation and Algorithms

Let $\mathcal{T} = (\mathcal{T}_i, i \in \mathcal{I})$ be a set of tasks available for meta-learning, where tasks are sampled for use by a distribution of $p_\mathcal{T}$. Each task $\mathcal{T}_i$ contains a training sample set $\mathcal{S}_i$ and a test set $\mathcal{D}_i$. Suppose that meta-learning divides all model parameters into mutually-exclusive sets $(w, \phi)$ as described below.

- $w$ includes task-specific parameters, and meta-learning trains a good initialization of $w$.
- $\phi$ includes common parameters shared by all tasks, and meta-learning trains $\phi$ for direct reuse.

For example, in training neural networks, $w$ often represents the parameters of some partial layers, and $\phi$ represents the parameters of the remaining inner layers. The goal of meta-learning here is to jointly learn $w$ as a good initialization parameter and $\phi$ as a reuse parameter, such that $(w_N, \phi)$ performs well on a sampled individual task $\mathcal{T}$, where $w_N$ is the $N$-step gradient descent update of $w$. To this end, ANIL solves the following optimization problem with the objective function given by

$$\text{(Meta objective function):} \quad \min_{w, \phi} L^{meta}(w, \phi) := \mathbb{E}_{i \sim p_\mathcal{T}} L_{\mathcal{D}_i}(w_N^i(w, \phi), \phi), \tag{1}$$

where the loss function $L_{\mathcal{D}_i}(w_N^i, \phi) := \sum_{\xi \in \mathcal{D}_i} \ell(w_N^i, \phi; \xi)$ takes the finite-sum form over the test dataset $\mathcal{D}_i$, and the parameter $w_N^i$ for task $i$ is obtained via an inner-loop $N$-step gradient descent update of $w_0^i = w$ (aiming to minimize the task $i$'s loss function $L_{\mathcal{S}_i}(w, \phi)$ over $w$) as given by

$$\text{(Inner-loop gradient descent):} \quad w_{m+1}^i = w_m^i - \alpha \nabla_w L_{\mathcal{S}_i}(w_m^i, \phi), \ m = 0, 1, ..., N-1. \tag{2}$$

Here, $w_N^i(w, \phi)$ explicitly indicates the dependence of $w_N^i$ on $\phi$ and the initialization $w$ via the iterative updates in eq. (2). To draw connection, the problem here reduces to the MAML [5] framework if $w$ includes all training parameters and $\phi$ is empty, i.e., no parameters are reused directly.

### 2.1 ANIL Algorithm

ANIL [24] (as described in Algorithm 1) solves the problem in eq. (1) via two nested optimization loops, i.e., inner loop for task-specific adaptation and outer loop for updating meta-initialization and reuse parameters. At the $k$-th outer loop, ANIL samples a batch $\mathcal{B}_k$ of identical and independently distributed (i.i.d.) tasks based on $p_\mathcal{T}$. Then, each task in $\mathcal{B}_k$ runs an inner loop of $N$ steps of gradient descent with a stepsize $\alpha$ as in lines 5-7 in Algorithm 1, where $w_{k,0}^i = w_k$ for all tasks $\mathcal{T}_i \in \mathcal{B}_k$.

After obtaining the inner-loop output $w_{k,N}^i$ for all tasks, ANIL computes two partial gradients $\frac{\partial L_{\mathcal{D}_i}(w_{k,N}^i, \phi_k)}{\partial w_k}$ and $\frac{\partial L_{\mathcal{D}_i}(w_{k,N}^i, \phi_k)}{\partial \phi_k}$ respectively by back-propagation, and updates $w_k$ and $\phi_k$ by

**Algorithm 1** ANIL Algorithm

---

1: **Input:** Distribution over tasks $p_{\mathcal{T}}$, inner stepsize $\alpha$, outer stepsize $\beta_w, \beta_\phi$, initialization $w_0, \phi_0$
2: **while** not converged **do**
3:     Sample a mini-batch of i.i.d. tasks $\mathcal{B}_k = \{\mathcal{T}_i\}_{i=1}^B$ based on the distribution $p_{\mathcal{T}}$
4:     **for** each task $\mathcal{T}_i$ in $\mathcal{B}_k$ **do**
5:         **for** $m = 0, 1, ..., N-1$ **do**
6:             Update $w_{k,m+1}^i = w_{k,m}^i - \alpha \nabla_w L_{\mathcal{S}_i}(w_{k,m}^i, \phi_k)$
7:         **end for**
8:         Compute gradients $\frac{\partial L_{\mathcal{D}_i}(w_{k,N}^i, \phi_k)}{\partial w_k}, \frac{\partial L_{\mathcal{D}_i}(w_{k,N}^i, \phi_k)}{\partial \phi_k}$ by back-propagation
9:     **end for**
10:    Update parameters $w_k$ and $\phi_k$ by mini-batch SGD:

$$w_{k+1} = w_k - \frac{\beta_w}{B} \sum_{i \in \mathcal{B}_k} \frac{\partial L_{\mathcal{D}_i}(w_{k,N}^i, \phi_k)}{\partial w_k}, \quad \phi_{k+1} = \phi_k - \frac{\beta_\phi}{B} \sum_{i \in \mathcal{B}_k} \frac{\partial L_{\mathcal{D}_i}(w_{k,N}^i, \phi_k)}{\partial \phi_k}$$

11:    Update $k \leftarrow k + 1$
12: **end while**

---

stochastic gradient descent as in line 10 in Algorithm 1. Note that $\phi_k$ and $w_k$ are treated to be mutually-independent during the differentiation process. Due to the nested dependence of $w_{k,N}^i$ on $\phi_k$ and $w_k$, the two partial gradients involve complicated second-order derivatives. Their explicit forms are provided in the following proposition.

**Proposition 1.** *The partial meta gradients take the following explicit form:*

1)    $\dfrac{\partial L_{\mathcal{D}_i}(w_{k,N}^i, \phi_k)}{\partial w_k} = \prod_{m=0}^{N-1} (I - \alpha \nabla_w^2 L_{\mathcal{S}_i}(w_{k,m}^i, \phi_k)) \nabla_w L_{\mathcal{D}_i}(w_{k,N}^i, \phi_k).$

2)    $\dfrac{\partial L_{\mathcal{D}_i}(w_{k,N}^i, \phi_k)}{\partial \phi_k} = - \alpha \sum_{m=0}^{N-1} \nabla_\phi \nabla_w L_{\mathcal{S}_i}(w_{k,m}^i, \phi_k) \prod_{j=m+1}^{N-1} (I - \alpha \nabla_w^2 L_{\mathcal{S}_i}(w_{k,j}^i, \phi_k)) \nabla_w L_{\mathcal{D}_i}(w_{k,N}^i, \phi_k)$
$\qquad\qquad\qquad\quad + \nabla_\phi L_{\mathcal{D}_i}(w_{k,N}^i, \phi_k).$

## 2.2    Technical Assumptions and Definitions

We let $z = (w, \phi) \in \mathbb{R}^n$ denote all parameters. For simplicity, suppose $\mathcal{S}_i$ and $\mathcal{D}_i$ for all $i \in \mathcal{I}$ have sizes of $S$ and $D$, respectively. In this paper, we consider the following types of loss functions.

- The outer-loop meta loss function in eq. (1) takes the finite-sum form as $L_{\mathcal{D}_i}(w_N^i, \phi) := \sum_{\xi \in \mathcal{D}_i} \ell(w_N^i, \phi; \xi)$. It is generally nonconvex in terms of both $w$ and $\phi$.

- The inner-loop loss function $L_{\mathcal{S}_i}(w, \phi)$ with respect to $w$ has two cases: strongly-convexity and nonconvexity. The strongly-convex case occurs often when $w$ corresponds to parameters of the last *linear* layer of a neural network, so that the loss function of such a $w$ is naturally chosen to be a quadratic function or a logistic loss with a strongly convex regularizer [3, 18]. The nonconvex case can occur if $w$ represents parameters of more than one layers (e.g., last two layers [24]). As we prove in Section 3, such geometries affect the convergence rate significantly.

Since the objective function $L^{meta}(w, \phi)$ in eq. (1) is generally nonconvex, we use the gradient norm as the convergence criterion, which is standard in nonconvex optimization.

**Definition 1.** *We say that $(\bar{w}, \bar{\phi})$ is an $\epsilon$-accurate solution for the meta optimization problem in eq.* (1) *if* $\mathbb{E} \left\| \frac{\partial L^{meta}(\bar{w}, \bar{\phi})}{\partial \bar{w}} \right\|^2 < \epsilon$ *and* $\mathbb{E} \left\| \frac{\partial L^{meta}(\bar{w}, \bar{\phi})}{\partial \bar{\phi}} \right\|^2 < \epsilon.$

We further take the following standard assumptions on the *individual* loss function for each task, which have been commonly adopted in conventional minimization problems [10, 13, 31] and min-max optimization [19] as well as the MAML-type optimization [7, 14].

**Assumption 1.** *The loss function $L_{\mathcal{S}_i}(z)$ and $L_{\mathcal{D}_i}(z)$ for each task $\mathcal{T}_i$ satisfy:*

- $L_{\mathcal{S}_i}(z)$ and $L_{\mathcal{D}_i}(z)$ are $L$-smooth, i.e., for any $z, z' \in \mathbb{R}^n$,
$$\|\nabla L_{\mathcal{S}_i}(z) - \nabla L_{\mathcal{S}_i}(z')\| \leq L\|z - z'\|, \|\nabla L_{\mathcal{D}_i}(z) - \nabla L_{\mathcal{D}_i}(z')\| \leq L\|z - z'\|.$$

- $L_{\mathcal{D}_i}(z)$ is $M$-Lipschitz, i.e., for any $z, z' \in \mathbb{R}^n$, $|L_{\mathcal{D}_i}(z) - L_{\mathcal{D}_i}(z')| \leq M\|z - z'\|$.

Note that we *do not* impose the function Lipschitz assumption (i.e., item 2 in Assumption 1) on the inner-loop loss function $L_{S_i}(z)$. We take the assumption on the Lipschitzness of function $L_{\mathcal{D}_i}$ to ensure the meta gradient to be bounded. We note that iMAML [25] alternatively assumes the search space of parameters to be bounded (see Theorem 1 therein) so that the meta gradient (eq. (5) therein) can be bounded.

As shown in Proposition 1, the partial meta gradients involve two types of high-order derivatives $\nabla_w^2 L_{\mathcal{S}_i}(\cdot, \cdot)$ and $\nabla_\phi \nabla_w L_{\mathcal{S}_i}(\cdot, \cdot)$. The following assumption imposes a Lipschitz condition for these two high-order derivatives, which has been widely adopted in optimization problems that involve two sets of parameters, e.g, bi-level programming [11].

**Assumption 2.** *Both $\nabla_w^2 L_{\mathcal{S}_i}(z)$ and $\nabla_\phi \nabla_w L_{\mathcal{S}_i}(z)$ are $\rho$-Lipschitz and $\tau$-Lipschitz, i.e.,*

- *For any $z, z' \in \mathbb{R}^n$, $\|\nabla_w^2 L_{\mathcal{S}_i}(z) - \nabla_w^2 L_{\mathcal{S}_i}(z')\| \leq \rho\|z - z'\|$.*

- *For any $z, z' \in \mathbb{R}^n$, $\|\nabla_\phi \nabla_w L_{\mathcal{S}_i}(z) - \nabla_\phi \nabla_w L_{\mathcal{S}_i}(z')\| \leq \tau\|z - z'\|$.*

## 3 Convergence Analysis of ANIL

We first provide convergence analysis for the ANIL algorithm, and then compare the performance of ANIL under two geometries and compare the performance between ANIL and MAML.

### 3.1 Convergence Analysis under Strongly-Convex Inner-Loop Geometry

We first analyze the convergence rate of ANIL for the case where the inner-loop loss function $L_{\mathcal{S}_i}(\cdot, \phi)$ satisfies the following strongly-convex condition.

**Definition 2.** *$L_{\mathcal{S}_i}(w, \phi)$ is $\mu$-strongly convex with respect to $w$ if for any $w, w'$ and $\phi$,*
$$L_{\mathcal{S}_i}(w', \phi) \geq L_{\mathcal{S}_i}(w, \phi) + \langle w' - w, \nabla_w L_{\mathcal{S}_i}(w, \phi)\rangle + \frac{\mu}{2}\|w - w'\|^2.$$

Based on Proposition 1, we characterize the smoothness property of $L^{meta}(w, \phi)$ in eq. (1) as below.

**Proposition 2.** *Suppose Assumptions 1 and 2 hold and choose the inner stepsize $\alpha = \frac{\mu}{L^2}$. Then, for any two points $(w_1, \phi_1), (w_2, \phi_2) \in \mathbb{R}^n$, we have*

1) $\left\| \frac{\partial L^{meta}(w, \phi)}{\partial w}\Big|_{(w_1, \phi_1)} - \frac{\partial L^{meta}(w, \phi)}{\partial w}\Big|_{(w_2, \phi_2)} \right\|$

$\leq \text{poly}(L, M, \rho)\frac{L}{\mu}(1 - \alpha\mu)^N\|w_1 - w_2\| + \text{poly}(L, M, \rho)\left(\frac{L}{\mu} + 1\right)N(1 - \alpha\mu)^N\|\phi_1 - \phi_2\|,$

2) $\left\| \frac{\partial L^{meta}(w, \phi)}{\partial \phi}\Big|_{(w_1, \phi_1)} - \frac{\partial L^{meta}(w, \phi)}{\partial \phi}\Big|_{(w_2, \phi_2)} \right\|$

$\leq \text{poly}(L, M, \tau, \rho)\frac{L}{\mu}(1 - \alpha\mu)^{\frac{N}{2}}\|w_1 - w_2\| + \text{poly}(L, M, \rho)\frac{L^3}{\mu^3}\|\phi_1 - \phi_2\|,$

*where $\tau, \rho, L$ and $M$ are given in Assumptions 1 and 2, and $\text{poly}(\cdot)$ denotes the polynomial function of the parameters with the explicit forms given in Appendix C.2.*

Proposition 2 indicates that increasing the number $N$ of inner-loop gradient descent steps yields much *smaller* smoothness parameters for the meta objective function $L^{meta}(w, \phi)$. As shown in the following theorem, this allows a larger stepsize $\beta_w$, which yields a faster convergence rate $\mathcal{O}(\frac{1}{K\beta_w})$.

**Theorem 1.** *Suppose Assumptions 1 and 2 hold, and apply Algorithm 1 to solve the meta optimization problem eq. (1) with stepsizes $\beta_w = \text{poly}(\rho, \tau, L, M)\mu^2(1 - \frac{\mu^2}{L^2})^{-\frac{N}{2}}$ and $\beta_\phi = \text{poly}(\rho, \tau, L, M)\mu^3$.*

*Then, ANIL finds a point $(w, \phi) \in \{(w_k, \phi_k), k = 0, ..., K-1\}$ such that*

$$\text{(Rate w.r.t. } w) \quad \mathbb{E}\left\|\frac{\partial L^{meta}(w,\phi)}{\partial w}\right\|^2 \leq \mathcal{O}\left(\frac{\frac{1}{\mu^2}\left(1-\frac{\mu^2}{L^2}\right)^{\frac{N}{2}}}{K} + \frac{\frac{1}{\mu}\left(1-\frac{\mu^2}{L^2}\right)^{\frac{N}{2}}}{B}\right),$$

$$\text{(Rate w.r.t. } \phi) \quad \mathbb{E}\left\|\frac{\partial L^{meta}(w,\phi)}{\partial \phi}\right\|^2 \leq \mathcal{O}\left(\frac{\frac{1}{\mu^2}\left(1-\frac{\mu^2}{L^2}\right)^{\frac{N}{2}} + \frac{1}{\mu^3}}{K} + \frac{\frac{1}{\mu}\left(1-\frac{\mu^2}{L^2}\right)^{\frac{3N}{2}} + \frac{1}{\mu^2}}{B}\right).$$

*To achieve an $\epsilon$-accurate point, ANIL requires at most $\mathcal{O}\left(\frac{c_w N}{\mu^4}\left(1-\frac{\mu^2}{L^2}\right)^{N/2} + \frac{c_w' N}{\mu^5}\right)\epsilon^{-2}$ gradient evaluations in $w$, $\mathcal{O}\left(\frac{c_\phi}{\mu^4}\left(1-\frac{\mu^2}{L^2}\right)^{N/2} + \frac{c_\phi'}{\mu^5}\right)\epsilon^{-2}$ gradient evaluations in $\phi$, and $\mathcal{O}\left(\frac{c_s N}{\mu^4}\left(1-\frac{\mu^2}{L^2}\right)^{N/2} + \frac{c_s' N}{\mu^5}\right)\epsilon^{-2}$ second-order derivative evaluations of $\nabla_w^2 L_{S_i}(\cdot, \cdot)$ and $\nabla_\phi \nabla_w L_{S_i}(\cdot, \cdot)$, where constants $c_w, c_w', c_\phi, c_\phi', c_s, c_s'$ depend on $\tau, M, \rho$.*

Theorem 1 shows that ANIL converges sublinearly with the number $K$ of outer-loop meta iterations, and the convergence error decays sublinearly with the number $B$ of sampled tasks, which are consistent with the nonconvex nature of the meta objective function. The convergence rate is further significantly affected by the number $N$ of the inner-loop steps. Specifically, with respect to $w$, ANIL converges exponentially fast as $N$ increases due to the strong convexity of the inner-loop loss. With respect to $\phi$, the convergence rate depends on two components: an exponential decay term with $N$ and an $N$-independent term. As a result, the overall convergence of meta optimization becomes faster as $N$ increases, and then saturates for large enough $N$ as the second component starts to dominate. This is demonstrated by our experiments in Section 4.1.

Theorem 1 further indicates that ANIL attains an $\epsilon$-accurate stationary point with the gradient and second-order evaluations at the order of $\mathcal{O}(\epsilon^{-2})$ due to nonconvexity of the meta objective function. The computational cost is further significantly affected by inner-loop steps. Specifically, the gradient and second-order derivative evaluations contain two terms: an exponential decay term with $N$ and a linear growth term with $N$. For a large condition number $\kappa$, a small $N$, e.g., $N = 2$, is a better choice. However, when $\kappa$ is not very large, e.g., in our experiments in Section 4.1 (in which increasing $N$ accelerates the iteration rate), the computational cost of ANIL initially decreases because the exponential reduction dominates the linear growth. But when $N$ is large enough, the exponential decay saturates and the linear growth dominates, and hence the overall computational cost of ANIL gets higher as $N$ further increases. This suggests to take a moderate but not too large $N$ in practice to achieve an optimized performance, which we also demonstrate in our experiments in Section 4.1.

### 3.2 Convergence Analysis under Nonconvex Inner-Loop Geometry

In this subsection, we study the case, in which the inner-loop loss function $L_{S_i}(\cdot, \phi)$ is nonconvex. The following proposition characterizes the smoothness of $L^{meta}(w, \phi)$ in eq. (1).

**Proposition 3.** *Suppose Assumptions 1 and 2 hold, and choose the inner-loop stepsize $\alpha < \mathcal{O}(\frac{1}{N})$. Then, for any two points $(w_1, \phi_1)$, $(w_2, \phi_2) \in \mathbb{R}^n$, we have*

$$1)\left\|\frac{\partial L^{meta}(w,\phi)}{\partial w}\Big|_{(w_1,\phi_1)} - \frac{\partial L^{meta}(w,\phi)}{\partial w}\Big|_{(w_2,\phi_2)}\right\| \leq \text{poly}(M,\rho,\alpha,L)N(\|w_1 - w_2\| + \|\phi_1 - \phi_2\|),$$

$$2)\left\|\frac{\partial L^{meta}(w,\phi)}{\partial \phi}\Big|_{(w_1,\phi_1)} - \frac{\partial L^{meta}(w,\phi)}{\partial \phi}\Big|_{(w_2,\phi_2)}\right\| \leq \text{poly}(M,\rho,\tau,\alpha,L)N(\|w_1 - w_2\| + \|\phi_1 - \phi_2\|),$$

*where $\tau, \rho, L$ and $M$ are given by Assumptions 1 and 2, and $\text{poly}(\cdot)$ denotes the polynomial function of the parameters with the explicit forms of the smoothness parameters given in Appendix D.1.*

Proposition 3 indicates that the meta objective function $L^{meta}(w, \phi)$ is smooth with respect to both $w$ and $\phi$ with their smoothness parameters increasing linearly with $N$. Hence, $N$ should be chosen to be small so that the outer-loop meta optimization can take reasonably large stepsize to run fast. Such a property is in sharp contrast to the strongly-convex case in which the corresponding smoothness parameters decrease with $N$.

The following theorem provides the convergence rate of ANIL under the nonconvex inner-loop loss.

**Theorem 2.** *Under the setting of Proposition 3, and apply Algorithm 1 to solve the meta optimization problem in eq. (1) with the stepsizes $\beta_w = \beta_\phi = \mathrm{poly}(\rho, \tau, M, \alpha, L)N^{-1}$. Then, ANIL finds a point $(w, \phi) \in \{(w_k, \phi_k), k = 0, ..., K - 1\}$ such that*

$$\mathbb{E}\left\|\frac{\partial L^{meta}(w, \phi)}{\partial w}\right\|^2 \leq \mathcal{O}\left(\frac{N}{K} + \frac{N}{B}\right), \qquad \mathbb{E}\left\|\frac{\partial L^{meta}(w, \phi)}{\partial \phi}\right\|^2 \leq \mathcal{O}\left(\frac{N}{K} + \frac{N}{B}\right).$$

*To achieve an $\epsilon$-accurate point, ANIL requires at most $\mathcal{O}(N^2\epsilon^{-2})$ gradient evaluations in $w$, $\mathcal{O}(N\epsilon^{-2})$ gradient evaluations in $\phi$, and $\mathcal{O}(N^2\epsilon^{-2})$ second-order derivative evaluations.*

Theorem 2 shows that ANIL converges sublinearly with $K$, the convergence error decays sublinearly with $B$, and the computational complexity scales at the order of $\mathcal{O}(\epsilon^{-2})$. But the nonconvexity of the inner loop affects the convergence very differently. Specifically, increasing the number $N$ of the inner-loop gradient descent steps yields slower convergence and higher computational complexity. This suggests to choose a relatively small $N$ for an efficient optimization process, which is demonstrated in our experiments in Section 4.2

### 3.3 Complexity Comparison of Different Geometries and Different Algorithms

In this subsection, we first compare the performance for ANIL under strongly convex and nonconvex inner-loop loss functions, and then compare the performance between ANIL and MAML.

Table 1: Comparison of different geometries on the convergence rate and complexity of ANIL.

| Geometries | Convergence rate | Gradient complexity | Second-order complexity |
|---|---|---|---|
| Strongly convex | $\mathcal{O}\left(\frac{(1-\xi)^{\frac{N}{2}}+c_k}{K} + \frac{(1-\xi)^{\frac{3N}{2}}+c_b}{B}\right)^\sharp$ | $\mathcal{O}\left(\frac{N((1-\xi)^{\frac{N}{2}}+c_\epsilon)}{\epsilon^2}\right)^\S$ | $\mathcal{O}\left(\frac{N((1-\xi)^{\frac{N}{2}}+c_\epsilon)}{\epsilon^2}\right)$ |
| Nonconvex | $\mathcal{O}\left(\frac{N}{K} + \frac{N}{B}\right)$ | $\mathcal{O}\left(\frac{N^2}{\epsilon^2}\right)$ | $\mathcal{O}\left(\frac{N^2}{\epsilon^2}\right)$ |

Each order term in the table summarizes the dominant components of both $w$ and $\phi$.
$\sharp: \xi = \frac{\mu^2}{L^2} < 1$, $c_k, c_b$ are constants. $\S: c_\epsilon$ is constant.

**Comparison for ANIL between strongly convex and nonconvex inner-loop geometries:** Our results in Sections 3.1 and 3.2 have showed that the inner-loop geometry can significantly affect the convergence rate and the computational complexity of ANIL. The detailed comparison is provided in Table 1. It can be seen that increasing $N$ yields a faster convergence rate for the strongly-convex inner loop, but a slower convergence rate for the nonconvex inner loop. Table 1 also indicates that increasing $N$ first reduces and then increases the computational complexity for the strongly-convex inner loop, but constantly increases the complexity for the nonconvex inner loop.

We next provide an intuitive explanation for such different behaviors under these two geometries. For the nonconvex inner loop, $N$ gradient descent iterations starting from two different initializations likely reach two points that are far away from each other due to the nonconvex landscape so that the meta objective function can have a large smoothness parameter. Consequently, the stepsize should be small to avoid divergence, which yields slow convergence. However, for the strongly-convex inner loop, also consider two $N$-step inner-loop gradient descent paths. Due to the strong convexity, they both approach to the same unique optimal point, and hence their corresponding values of the meta objective function are guaranteed to be close to each other as $N$ increases. Thus, increasing $N$ reduces the smoothness parameter, and allows a faster convergence rate.

**Comparison between ANIL and MAML:** [24] empirically showed that ANIL significantly speeds up MAML due to the fact that only a very small subset of parameters go through the inner-loop update. The complexity results in Theorem 1 and Theorem 2 provide theoretical characterization of such an acceleration. To formally compare the performance between ANIL and MAML, let $n_w$ and $n_\phi$ be the dimensions of $w$ and $\phi$, respectively. The detailed comparison is provided in Table 2.

For ANIL with the strongly-convex inner loop, Table 2 shows that ANIL requires fewer gradient and second-order entry evaluations than MAML by a factor of $\mathcal{O}\left(\frac{Nn_w+Nn_\phi}{Nn_w+n_\phi}(1+\kappa L)^N\right)$ and $\mathcal{O}\left(\frac{n_w+n_\phi}{n_w}(1+\kappa L)^N\right)$, respectively. Such improvements are significant because $n_\phi$ is often much larger than $n_w$.

Table 2: Comparison of the computational complexities of ANIL and MAML.

| Algorithms | # of gradient entry evaluations $^\sharp$ | # of second-order entry evaluations$^\S$ |
|---|---|---|
| MAML [14, Theorem 2] | $\mathcal{O}\left(\frac{(Nn_w+Nn_\phi)(1+\kappa L)^N}{\epsilon^2}\right)$$^\aleph$ | $\mathcal{O}\left(\frac{(n_w+n_\phi)^2 N(1+\kappa L)^N}{\epsilon^2}\right)$ |
| ANIL (Strongly convex) | $\mathcal{O}\left(\frac{(Nn_w+n_\phi)((1-\xi)^{\frac{N}{2}}+c_\epsilon)}{\epsilon^2}\right)$$^\flat$ | $\mathcal{O}\left(\frac{(n_w^2+n_w n_\phi)N((1-\xi)^{\frac{N}{2}}+c_\epsilon)}{\epsilon^2}\right)$ |
| ANIL (Nonconvex) | $\mathcal{O}\left(\frac{(Nn_w+n_\phi)N}{\epsilon^2}\right)$ | $\mathcal{O}\left(\frac{(n_w^2+n_w n_\phi)N^2}{\epsilon^2}\right)$ |

$\sharp$: with respect to each dimension of gradient. $\S$: with respect to each entry of second-order derivatives.
$\aleph$: $\kappa$ is the inner-loop stepsize used in MAML. $\flat$ : $\xi = \frac{\mu^2}{L^2} < 1$ and $c_\epsilon$ is a constant.

For nonconvex inner loop, we set $\kappa \le 1/N$ for MAML [14, Corollary 2] to be consistent with our analysis for ANIL in Theorem 2. Then, Table 2 indicates that ANIL requires fewer gradient and second-order entry computations than MAML by a factor of $\mathcal{O}\left(\frac{Nn_w+Nn_\phi}{Nn_w+n_\phi}\right)$ and $\mathcal{O}\left(\frac{n_w+n_\phi}{n_w}\right)$.

## 4   Experiments

In this section, we validate our theory on the ANIL algorithm over two benchmarks for few-shot multiclass classification, i.e., FC100 [23] and miniImageNet [30]. The experimental implementation and the model architectures are adapted from the existing repository [1] for ANIL. We consider a 5-way 5-shot task on both the FC100 and miniImageNet datasets. We relegate the introduction of datasets, model architectures and hyper-parameter settings to Appendix A due to the space limitations.

Our experiments aim to explore how the different geometry (i.e., strong convexity and nonconvexity) of the inner loop affects the convergence performance of ANIL.

### 4.1   ANIL with Strongly-Convex Inner-Loop Loss

We first validate the convergence results of ANIL under the *strongly-convex* inner-loop loss function $L_{\mathcal{S}_i}(\cdot, \phi)$, as we establish in Section 3.1. Here, we let $w$ be parameters of *the last layer* of CNN and $\phi$ be parameters of the remaining inner layers. As in [3, 18], the inner-loop loss function adopts $L^2$ regularization on $w$ with a hyper-parameter $\lambda > 0$, and hence is *strongly convex*.

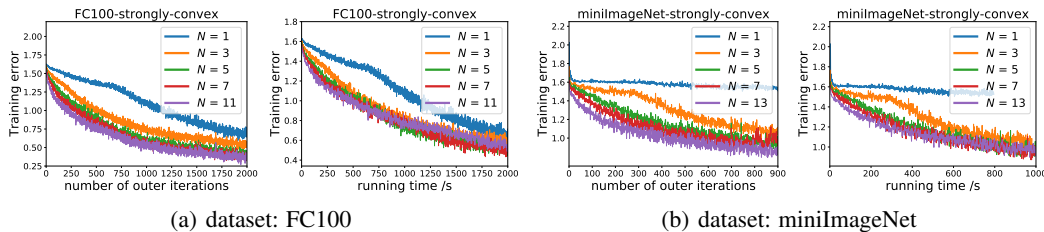

(a) dataset: FC100    (b) dataset: miniImageNet

Figure 1: Convergence of ANIL with strongly-convex inner-loop loss function. For each dataset, left plot: training loss v.s. number of total meta iterations; right plot: training loss v.s. running time.

For the FC100 dataset, the left plot of Figure 1(a) shows that the convergence rate in terms of the number of meta outer-loop iterations becomes faster as the inner-loop steps $N$ increases, but nearly saturates at $N = 7$ (i.e., there is not much improvement for $N \ge 7$). This is consistent with Theorem 1, in which the gradient convergence bound first decays exponentially with $N$, and then the bound in $\phi$ dominates and saturates to a constant. Furthermore, the right plot of Figure 1(a) shows that the running-time convergence first becomes faster as $N$ increases up to $N \le 7$, and then starts to slow down as $N$ further increases. This is also captured by Theorem 1 as follows. The computational cost of ANIL initially decreases because the exponential reduction dominates the linear growth in the gradient and second-order derivative evaluations. But when $N$ becomes large enough, the linear growth dominates, and hence the overall computational cost of ANIL gets higher as $N$ further increases. Similar nature of convergence behavior is also observed over the miniImageNet dataset as shown in Figure 1(b). Thus, our experiment suggests that for the strongly-convex inner-loop loss,

choosing a relatively large $N$ (e.g., $N = 7$) achieves a good balance between the convergence rate (as well as the convergence error) and the computational complexity.

## 4.2 ANIL with Nonconvex Inner-Loop Loss

We next validate the convergence results of ANIL under the *nonconvex* inner-loop loss function $L_{\mathcal{S}_i}(\cdot, \phi)$, as we establish in Section 3.2. Here, we let $w$ be the parameters of *the last two layers with ReLU activation* of CNN (and hence the inner-loop loss is nonconvex with respect to $w$) and $\phi$ be the remaining parameters of the inner layers.

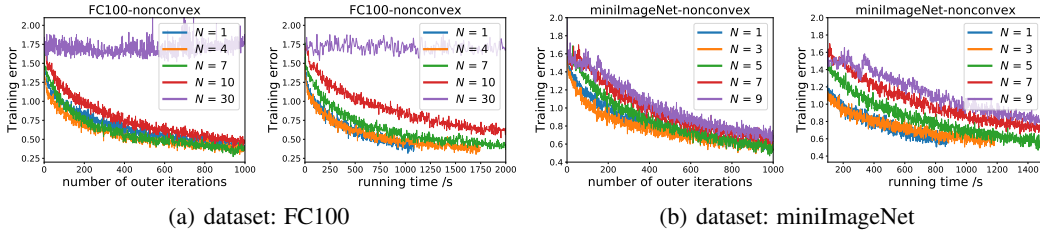

(a) dataset: FC100                                     (b) dataset: miniImageNet

Figure 2: Convergence of ANIL with nonconvex inner-loop loss function. For each dataset, left plot: training loss v.s. number of total meta iterations; right plot: training loss v.s. running time.

Figure 2 provides the experimental results over the datasets FC100 and miniImageNet. For both datasets, the running-time convergence (right plot for each dataset) becomes *slower* as $N$ increases, where $N = 1$ is fastest, and the algorithm even diverges for $N = 30$ over the FC100 dataset. The plots are consist with Theorem 2, in which the computational complexity increases as $N$ becomes large. Note that $N = 1$ is not the fastest in the left plot for each dataset because the influence of $N$ is more prominent in terms of the running time than the number of outer-loop iterations (which is likely offset by other constant-level parameters for small $N$). Thus, the optimization perspective here suggests that $N$ should be chosen as small as possible for computational efficiency, which in practice should be jointly considered with other aspects such as generalization for determining $N$.

## 5 Conclusion

In this paper, we provide theoretical convergence guarantee for the ANIL algorithm under strongly-convex and nonconvex inner-loop loss functions, respectively. Our analysis reveals different performance behaviors of ANIL under the two geometries by characterizing the impact of inner-loop adaptation steps on the overall convergence rate. Our results further provide guidelines for the hyper-parameter selections for ANIL under different inner-loop loss geometries.

## Broader Impact

Meta-learning has been successfully used in a wide range of applications including reinforcement learning, robotics, federated learning, imitation learning, etc, which will be highly influential to technologize our life. This work focuses on understanding the computational efficiency of the optimization-based meta learning algorithms, particularly MAML and ANIL type algorithms. We characterize the convergence guarantee on these algorithms. Furthermore, our theory provides useful guidelines on the selections of hyperparameters for these algorithms, in order for them to be efficiently implemented in large-scale applications. We also anticipate the theory that we develop will be useful in other academic fields in addition to machine learning, including optimization theory, signal processing, and statistics.

## Acknowledgments and Disclosure of Funding

The work of K. Ji and Y. Liang is supported in part by the U.S. National Science Foundation under the grants CCF-1900145 and CCF-1761506. J. D. Lee acknowledges support of NSF CCF 2002272. H. Vincent Poor acknowledges support of the U.S. National Foundation under Grant CCF-1908308

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
