[Supplementary Material]

# Supplementary Materials

## A  Further Specification of Experiments

Following [1], we consider a 5-way 5-shot task on both the FC100 and miniImageNet datasets, where we evaluate the model's ability to discriminate 5 unseen classes, given only 5 labelled samples per class. We adopt Adam [15] as the optimizer for the meta outer-loop update, and adopt the cross-entropy loss to measure the error between the predicted and true labels.

### A.1  Introduction of FC100 and miniImageNet datasets

**FC100 dataset.** The FC100 dataset [23] is generated from CIFAR-100 [17], and consists of 100 classes with each class containing 600 images of size 32. Following recent work [23, 18], we split these 100 classes into 60 classes for meta-training, 20 classes for meta-validation, and 20 classes for meta-testing.

**miniImageNet dataset.** The miniImageNet dataset [30] consists of 100 classes randomly chosen from ImageNet [27], where each class contains 600 images of size $84 \times 84$. Following the repository [1], we partition these classes into 64 classes for meta-training, 16 classes for meta-validation, and 20 classes for meta-testing.

### A.2  Model Architectures and Hyper-Parameter Setting

We adopt the following four model architectures depending on the dataset and the geometry of the inner-loop loss. The hyper-parameter configuration for each architecture is also provided as follows.

**Case 1: FC100 dataset, strongly-convex inner-loop loss.** Following [1], we use a 4-layer CNN of four convolutional blocks, where each block sequentially consists of a $3 \times 3$ convolution with a padding of 1 and a stride of 2, batch normalization, ReLU activation, and $2 \times 2$ max pooling. Each convolutional layer has 64 filters. This model is trained with an inner-loop stepsize of 0.005, an outer-loop (meta) stepsize of 0.001, and a mini-batch size of $B = 32$. We set the regularization parameter $\lambda$ of the $L^2$ regularizer to be $\lambda = 5$.

**Case 2: FC100 dataset, nonconvex inner-loop loss.** We adopt a 5-layer CNN with the first four convolutional layers the same as in **Case 1**, followed by ReLU activation, and a full-connected layer with size of $256 \times$ ways. This model is trained with an inner-loop stepsize of 0.04, an outer-loop (meta) stepsize of 0.003, and a mini-batch size of $B = 32$.

**Case 3: miniImageNet dataset, strongly-convex inner-loop loss.** Following [24], we use a 4-layer CNN of four convolutional blocks, where each block sequentially consists of a $3 \times 3$ convolution with 32 filters, batch normalization, ReLU activation, and $2 \times 2$ max pooling. We choose an inner-loop stepsize of 0.002, an outer-loop (meta) stepsize of 0.002, and a mini-batch size of $B = 32$, and set the regularization parameter $\lambda$ of the $L^2$ regularizer to be $\lambda = 0.1$.

**Case 4: miniImageNet dataset, nonconvex inner-loop loss.** We adopt a 5-layer CNN with the first four convolutional layers the same as in **Case 3**, followed by ReLU activation, and a full-connected layer with size of $128 \times$ ways. We choose an inner-loop stepsize of 0.02, an outer-loop (meta) stepsize of 0.003, and a mini-batch size of $B = 32$.

### A.3  Experiments with SGD Optimizer

The experiments in Section 4.1 and Section 4.2 adopt the Adam optimizer. In this subsection, we conduct experiments using mini-batch stochastic gradient descent (SGD) on FC100 dataset. For both the strongly-convex and nonconvex cases, we choose an inner-loop stepsize of 0.05, an outer-loop (meta) stepsize of 0.05, and a mini-batch size of $B = 32$. The results are given in Figure 3. It can be seen that the nature of the results remains the same as those done with the Adam optimizer.

### A.4  Experiments on Comparison of ANIL and MAML

In Figure 4, we compare the computational efficiency between ANIL and MAML. For the miniImageNet dataset, we choose the inner-loop stepsize as 0.1, the outer-loop (meta) stepsize as 0.002, the

Figure 3: Convergence of ANIL with mini-batch SGD over FC100 dataset. Left plot: strongly-convex inner-loop loss; right plot: nonconvex inner-loop loss.

mini-batch size as 32, and the number of inner-loop steps as 5 for ANIL. For MAML, we choose the inner-loop stepsize as 0.5, the outer-loop stepsize as 0.003, the mini-batch size as 32, and the number of inner-loop steps as 3. For the FC100 dataset, we choose the inner-loop stepsize as 0.1, the outer-loop (meta) stepsize as 0.001, the mini-batch size as 32 for ANIL. For MAML, we choose the inner-loop stepsize as 0.5, the outer-loop stepsize as 0.001, and the mini-batch size as 32. We choose the number of inner-loop steps as 10 for ANIL and 3 for MAML. It can be seen that ANIL converges faster than MAML, as well supported by our theoretical results.

(a) dataset: FC100

(b) dataset: miniImageNet

Figure 4: Computational comparison of ANIL and MAML. For each dataset, left plot: training accuracy v.s. running time; right plot: test accuracy v.s. running time.

# B Proof of Proposition 1

We first prove the form of the partial gradient $\frac{\partial L_{\mathcal{D}_i}(w_{k,N}^i, \phi_k)}{\partial w_k}$. Using the chain rule, we have

$$
\begin{aligned}
\frac{\partial L_{\mathcal{D}_i}(w_{k,N}^i, \phi_k)}{\partial w_k} &= \frac{\partial w_{k,N}^i(w_k, \phi_k)}{\partial w_k} \nabla_w L_{\mathcal{D}_i}(w_{k,N}^i, \phi_k) + \frac{\partial \phi_k}{\partial w_k} \nabla_\phi L_{\mathcal{D}_i}(w_{k,N}^i, \phi_k) \\
&= \frac{\partial w_{k,N}^i(w_k, \phi_k)}{\partial w_k} \nabla_w L_{\mathcal{D}_i}(w_{k,N}^i, \phi_k),
\end{aligned}
\tag{3}
$$

where the last equality follows from the fact that $\frac{\partial \phi_k}{\partial w_k} = 0$. Recall that the gradient updates in Algorithm 1 are given by

$$
w_{k,m+1}^i = w_{k,m}^i - \alpha \nabla_w L_{\mathcal{S}_i}(w_{k,m}^i, \phi_k), \ m = 0, 1, ..., N-1,
\tag{4}
$$

where $w_{k,0}^i = w_k$ for all $i$. Taking derivatives w.r.t. $w_k$ in eq. (4) yields

$$
\frac{\partial w_{k,m+1}^i}{\partial w_k} = \frac{\partial w_{k,m}^i}{\partial w_k} - \alpha \frac{\partial w_{k,m}^i}{\partial w_k} \nabla_w^2 L_{\mathcal{S}_i}(w_{k,m}^i, \phi_k) - \alpha \underbrace{\frac{\partial \phi_k}{\partial w_k} \nabla_\phi \nabla_w L_{\mathcal{S}_i}(w_{k,m}^i, \phi_k)}_{0}.
\tag{5}
$$

Telescoping eq. (5) over $m$ from 0 to $N-1$ yields

$$
\frac{\partial w_{k,N}^i}{\partial w_k} = \prod_{m=0}^{N-1} (I - \alpha \nabla_w^2 L_{\mathcal{S}_i}(w_{k,m}^i, \phi_k)),
$$

which, in conjunction eq. (3), yields the first part in Proposition 1.

For the second part, using chain rule, we have

$$\frac{\partial L_{\mathcal{D}_i}(w_{k,N}^i, \phi_k)}{\partial \phi_k} = \frac{\partial w_{k,N}^i}{\partial \phi_k} \nabla_w L_{\mathcal{D}_i}(w_{k,N}^i, \phi_k) + \nabla_\phi L_{\mathcal{D}_i}(w_{k,N}^i, \phi_k). \tag{6}$$

Taking derivates w.r.t. $\phi_k$ in eq. (4) yields

$$\begin{aligned}
\frac{\partial w_{k,m+1}^i}{\partial \phi_k} =& \frac{\partial w_{k,m}^i}{\partial \phi_k} - \alpha \Big( \frac{\partial w_{k,m}^i}{\partial \phi_k} \nabla_w^2 L_{\mathcal{S}_i}(w_{k,m}^i, \phi_k) + \nabla_\phi \nabla_w L_{\mathcal{S}_i}(w_{k,m}^i, \phi_k) \Big) \\
=& \frac{\partial w_{k,m}^i}{\partial \phi_k} (I - \alpha \nabla_w^2 L_{\mathcal{S}_i}(w_{k,m}^i, \phi_k)) - \alpha \nabla_\phi \nabla_w L_{\mathcal{S}_i}(w_{k,m}^i, \phi_k).
\end{aligned}$$

Telescoping the above equality over $m$ from 0 to $N-1$ yields

$$\begin{aligned}
\frac{\partial w_{k,N}^i}{\partial \phi_k} = & \frac{\partial w_{k,0}^i}{\partial \phi_k} \prod_{m=0}^{N-1} (I - \alpha \nabla_w^2 L_{\mathcal{S}_i}(w_{k,m}^i, \phi_k)) \\
& - \alpha \sum_{m=0}^{N-1} \nabla_\phi \nabla_w L_{\mathcal{S}_i}(w_{k,m}^i, \phi_k) \prod_{j=m+1}^{N-1} (I - \alpha \nabla_w^2 L_{\mathcal{S}_i}(w_{k,j}^i, \phi_k)),
\end{aligned}$$

which, in conjunction with the fact that $\frac{\partial w_{k,0}^i}{\partial \phi_k} = \frac{\partial w_k}{\partial \phi_k} = 0$ and eq. (6), yields the second part.

## C  Proof in Section 3.1: Strongly-Convex Inner Loop

### C.1  Auxiliary Lemma

The following lemma characterizes a bound on the difference between $w_t^i(w_1, \phi_1)$ and $w_t^i(w_2, \phi_2)$, where $w_t^i(w, \phi)$ corresponds to the $t^{th}$ inner-loop iteration starting from the initialization point $(w, \phi)$.

**Lemma 1.** *Choose $\alpha$ such that $1 - 2\alpha\mu + \alpha^2 L^2 > 0$. Then, for any two points $(w_1, \phi_1), (w_2, \phi_2) \in \mathbb{R}^n$, we have*

$$\left\| w_t^i(w_1, \phi_1) - w_t^i(w_2, \phi_2) \right\| \leq (1 - 2\alpha\mu + \alpha^2 L^2)^{\frac{t}{2}} \|w_1 - w_2\| + \frac{\alpha L \|\phi_1 - \phi_2\|}{1 - \sqrt{1 - 2\alpha\mu + \alpha^2 L^2}}.$$

*Proof.* Based on the updates in eq. (2), we have

$$\begin{aligned}
w_{m+1}^i(w_1, \phi_1) - w_{m+1}^i(w_2, \phi_2) =& w_m^i(w_1, \phi_1) - w_m^i(w_2, \phi_2) \\
& - \alpha \big( \nabla_w L_{\mathcal{S}_i}(w_m^i(w_1, \phi_1), \phi_1) - \nabla_w L_{\mathcal{S}_i}(w_m^i(w_2, \phi_2), \phi_1) \big) \\
& + \alpha \big( \nabla_w L_{\mathcal{S}_i}(w_m^i(w_2, \phi_2), \phi_2) - \nabla_w L_{\mathcal{S}_i}(w_m^i(w_2, \phi_2), \phi_1) \big),
\end{aligned}$$

which, together with the triangle inequality and Assumption 1, yields

$$\begin{aligned}
& \|w_{m+1}^i(w_1, \phi_1) - w_{m+1}^i(w_2, \phi_2)\| \\
\leq & \underbrace{\left\| w_m^i(w_1, \phi_1) - w_m^i(w_2, \phi_2) - \alpha \big( \nabla_w L_{\mathcal{S}_i}(w_m^i(w_1, \phi_1), \phi_1) - \nabla_w L_{\mathcal{S}_i}(w_m^i(w_2, \phi_2), \phi_1) \big) \right\|}_{P} \\
& + \alpha L \|\phi_1 - \phi_2\|. \tag{7}
\end{aligned}$$

Our next step is to upper-bound the term $P$ in eq. (7). Note that

$$\begin{aligned}
P^2 =& \|w_m^i(w_1, \phi_1) - w_m^i(w_2, \phi_2)\|^2 + \alpha^2 \|\nabla_w L_{\mathcal{S}_i}(w_m^i(w_1, \phi_1), \phi_1) - \nabla_w L_{\mathcal{S}_i}(w_m^i(w_2, \phi_2), \phi_1)\|^2 \\
& - 2\alpha \Big\langle w_m^i(w_1, \phi_1) - w_m^i(w_2, \phi_2), \nabla_w L_{\mathcal{S}_i}(w_m^i(w_1, \phi_1), \phi_1) - \nabla_w L_{\mathcal{S}_i}(w_m^i(w_2, \phi_2), \phi_1) \Big\rangle \\
\leq & (1 + \alpha^2 L^2 - 2\alpha\mu) \|w_m^i(w_1, \phi_1) - w_m^i(w_2, \phi_2)\|^2, \tag{8}
\end{aligned}$$

where the last inequality follows from the strong-convexity of the loss function $L_{\mathcal{S}_i}(\cdot, \phi)$ that for any $w, w'$ and $\phi$,

$$\langle w - w', \nabla_w L_{\mathcal{S}_i}(w, \phi) - \nabla_w L_{\mathcal{S}_i}(w', \phi) \rangle \geq \mu \|w - w'\|^2.$$

Substituting eq. (8) into eq. (7) yields

$$\|w^i_{m+1}(w_1, \phi_1) - w^i_{m+1}(w_2, \phi_2)\| \leq \sqrt{1 + \alpha^2 L^2 - 2\alpha\mu} \|w^i_m(w_1, \phi_1) - w^i_m(w_2, \phi_2)\|$$
$$+ \alpha L \|\phi_1 - \phi_2\|. \tag{9}$$

Telescoping the above inequality over $m$ from 0 to $t - 1$ completes the proof. $\qquad\square$

## C.2 Proof of Proposition 2

Using an approach similar to the proof of Proposition 1, we have

$$\frac{\partial L_{\mathcal{D}_i}(w^i_N, \phi)}{\partial w} = \prod_{m=0}^{N-1} (I - \alpha \nabla^2_w L_{\mathcal{S}_i}(w^i_m, \phi)) \nabla_w L_{\mathcal{D}_i}(w^i_N, \phi). \tag{10}$$

Let $w^i_m(w, \phi)$ denote the $m^{th}$ inner-loop iteration starting from $(w, \phi)$. Then, we have

$$\left\| \frac{\partial L_{\mathcal{D}_i}(w^i_N, \phi)}{\partial w} \right|_{(w_1, \phi_1)} - \frac{\partial L_{\mathcal{D}_i}(w^i_N, \phi)}{\partial w} \right|_{(w_2, \phi_2)} \right\|$$

$$\leq \underbrace{\left\| \prod_{m=0}^{N-1} (I - \alpha \nabla^2_w L_{\mathcal{S}_i}(w^i_m(w_2, \phi_2), \phi_2)) \right\| \left\| \nabla_w L_{\mathcal{D}_i}(w^i_N(w_1, \phi_1), \phi_1) - \nabla_w L_{\mathcal{D}_i}(w^i_N(w_2, \phi_2), \phi_2) \right\|}_{P}$$

$$+ \left\| \prod_{m=0}^{N-1} (I - \alpha \nabla^2_w L_{\mathcal{S}_i}(w^i_m(w_1, \phi_1), \phi_1)) \nabla_w L_{\mathcal{D}_i}(w^i_N(w_1, \phi_1), \phi_1) \right.$$

$$\underbrace{\left. - \prod_{m=0}^{N-1} (I - \alpha \nabla^2_w L_{\mathcal{S}_i}(w^i_m(w_2, \phi_2), \phi_2)) \nabla_w L_{\mathcal{D}_i}(w^i_N(w_1, \phi_1), \phi_1) \right\|}_{Q}, \tag{11}$$

where $w^i_m(w, \phi)$ is obtained through the following gradient descent steps

$$w^i_{t+1}(w, \phi) = w^i_t(w, \phi) - \alpha \nabla_w L_{\mathcal{S}_i}(w^i_t(w, \phi), \phi), \ t = 0, ..., m - 1 \text{ and } w^i_0(w, \phi) = w. \tag{12}$$

We next upper-bound the term $P$ in eq. (11). Based on the strongly-convexity of the function $L_{\mathcal{S}_i}(\cdot, \phi)$, we have $\left\| I - \alpha \nabla^2_w L_{\mathcal{S}_i}(\cdot, \phi) \right\| \leq 1 - \alpha\mu$, and hence

$$P \leq (1 - \alpha\mu)^N \left\| \nabla_w L_{\mathcal{D}_i}(w^i_N(w_1, \phi_1), \phi_1) - \nabla_w L_{\mathcal{D}_i}(w^i_N(w_2, \phi_2), \phi_2) \right\|$$

$$\overset{(i)}{\leq} (1 - \alpha\mu)^N L \left( \|w^i_N(w_1, \phi_1) - w^i_N(w_2, \phi_2)\| + \|\phi_1 - \phi_2\| \right)$$

$$\overset{(ii)}{\leq} (1 - \alpha\mu)^N L \left( (1 - 2\alpha\mu + \alpha^2 L^2)^{\frac{N}{2}} \|w_1 - w_2\| + \frac{\alpha L \|\phi_1 - \phi_2\|}{1 - \sqrt{1 - 2\alpha\mu + \alpha^2 L^2}} + \|\phi_1 - \phi_2\| \right)$$

$$\overset{(iii)}{\leq} (1 - \alpha\mu)^{\frac{3N}{2}} L \|w_1 - w_2\| + (1 - \alpha\mu)^N L \left( \frac{2L}{\mu} + 1 \right) \|\phi_1 - \phi_2\|, \tag{13}$$

where $(i)$ follows from Assumption 1, (ii) follows from Lemma 1, and $(iii)$ follows from the fact that $\alpha\mu = \frac{\mu^2}{L^2} = \alpha^2 L^2$ and $\sqrt{1 - x} \leq 1 - \frac{1}{2}x$.

To upper-bound the term $Q$ in eq. (11), we have

$$Q \leq M \underbrace{\left\| \prod_{m=0}^{N-1} (I - \alpha \nabla^2_w L_{\mathcal{S}_i}(w^i_m(w_1, \phi_1), \phi_1)) - \prod_{m=0}^{N-1} (I - \alpha \nabla^2_w L_{\mathcal{S}_i}(w^i_m(w_2, \phi_2), \phi_2)) \right\|}_{P_{N-1}}. \tag{14}$$

To upper-bound $P_{N-1}$ in eq. (14), we define a more general quantity $P_t$ by replacing $N - 1$ with $t$ in eq. (14). Using the triangle inequality, we have

$$P_t \leq \alpha(1 - \alpha\mu)^t \|\nabla^2_w L_{\mathcal{S}_i}(w^i_t(w_1, \phi_1), \phi_1)) - \nabla^2_w L_{\mathcal{S}_i}(w^i_t(w_2, \phi_2), \phi_2))\| + (1 - \alpha\mu) P_{t-1}$$

$$\leq (1 - \alpha\mu) P_{t-1} + \alpha\rho(1 - \alpha\mu)^{\frac{3t}{2}} \|w_1 - w_2\| + (1 - \alpha\mu)^t \alpha\rho \left( \frac{2L}{\mu} + 1 \right) \|\phi_1 - \phi_2\|. \tag{15}$$

Telescoping eq. (15) over $t$ from 1 to $N-1$ yields

$$P_{N-1} \leq (1-\alpha\mu)^{N-1} P_0 + \sum_{t=1}^{N-1} \alpha\rho(1-\alpha\mu)^{\frac{3t}{2}} \|w_1 - w_2\|(1-\alpha\mu)^{N-1-t}$$

$$+ \sum_{t=1}^{N-1} (1-\alpha\mu)^t \alpha\rho \left(\frac{2L}{\mu} + 1\right) \|\phi_1 - \phi_2\|(1-\alpha\mu)^{N-1-t},$$

which, in conjunction with $P_0 \leq \alpha\rho(\|w_1 - w_2\| + \|\phi_1 - \phi_2\|)$, yields

$$P_{N-1} \leq (1-\alpha\mu)^{N-1} \alpha\rho(\|w_1 - w_2\| + \|\phi_1 - \phi_2\|) + \alpha\rho\|w_1 - w_2\|(1-\alpha\mu)^{N-1} \frac{\sqrt{1-\alpha\mu}}{1-\sqrt{1-\alpha\mu}}$$

$$+ \alpha\rho\left(\frac{2L}{\mu} + 1\right)\|\phi_1 - \phi_2\|(N-1)(1-\alpha\mu)^{N-1}$$

$$\leq \frac{2\rho}{\mu}(1-\alpha\mu)^{N-1}\|w_1 - w_2\| + \alpha\rho\left(\frac{2L}{\mu} + 1\right)\|\phi_1 - \phi_2\|N(1-\alpha\mu)^{N-1},$$

which, in conjunction with eq. (14), yields

$$Q \leq \frac{2\rho M}{\mu}(1-\alpha\mu)^{N-1}\|w_1 - w_2\| + \alpha\rho M\left(\frac{2L}{\mu} + 1\right)\|\phi_1 - \phi_2\|N(1-\alpha\mu)^{N-1}. \qquad (16)$$

Substituting eq. (13) and eq. (16) into eq. (11) yields

$$\left\|\frac{\partial L_{\mathcal{D}_i}(w_N^i, \phi)}{\partial w}\Big|_{(w_1, \phi_1)} - \frac{\partial L_{\mathcal{D}_i}(w_N^i, \phi)}{\partial w}\Big|_{(w_2, \phi_2)}\right\|$$

$$\leq \left((1-\alpha\mu)^{\frac{3N}{2}} L + \frac{2\rho M}{\mu}(1-\alpha\mu)^{N-1}\right)\|w_1 - w_2\|$$

$$+ \left((1-\alpha\mu)^N L + \alpha\rho M N(1-\alpha\mu)^{N-1}\right)\left(\frac{2L}{\mu} + 1\right)\|\phi_1 - \phi_2\|. \quad (17)$$

Based on the definition $L^{meta}(w, \phi) = \mathbb{E}_i L_{\mathcal{D}_i}(w_N^i, \phi)$ and using the Jensen's inequality, we have

$$\left\|\frac{\partial L^{meta}(w, \phi)}{\partial w}\Big|_{(w_1, \phi_1)} - \frac{\partial L^{meta}(w, \phi)}{\partial w}\Big|_{(w_2, \phi_2)}\right\|$$

$$\leq \mathbb{E}_i\left\|\frac{\partial L_{\mathcal{D}_i}(w_N^i, \phi)}{\partial w}\Big|_{(w_1, \phi_1)} - \frac{\partial L_{\mathcal{D}_i}(w_N^i, \phi)}{\partial w}\Big|_{(w_2, \phi_2)}\right\|. \quad (18)$$

Combining eq. (17) and eq. (18) completes the proof of the first item.

We next prove the Lipschitz property of the partial gradient $\frac{\partial L_{\mathcal{D}_i}(w_N^i, \phi)}{\partial \phi}$. For notational convenience, we define several quantities below.

$$Q_m(w, \phi) = \nabla_\phi \nabla_w L_{\mathcal{S}_i}(w_m^i(w, \phi), \phi), \ U_m(w, \phi) = \prod_{j=m+1}^{N-1}(I - \alpha\nabla_w^2 L_{\mathcal{S}_i}(w_j^i(w, \phi), \phi)),$$

$$V_m(w, \phi) = \nabla_w L_{\mathcal{D}_i}(w_N^i(w, \phi), \phi), \qquad (19)$$

where we let $w_m^i(w, \phi)$ denote the $m^{th}$ inner-loop iteration starting from $(w, \phi)$. Using an approach similar to the proof for Proposition 1, we have

$$\frac{\partial L_{\mathcal{D}_i}(w_N^i, \phi)}{\partial \phi} = -\alpha \sum_{m=0}^{N-1} \nabla_\phi \nabla_w L_{\mathcal{S}_i}(w_m^i, \phi) \prod_{j=m+1}^{N-1}(I - \alpha\nabla_w^2 L_{\mathcal{S}_i}(w_j^i, \phi))\nabla_w L_{\mathcal{D}_i}(w_N^i, \phi)$$

$$+ \nabla_\phi L_{\mathcal{D}_i}(w_N^i, \phi). \qquad (20)$$

Then, we have

$$\left\|\frac{\partial L_{\mathcal{D}_i}(w_N^i, \phi)}{\partial \phi}\Big|_{(w_1, \phi_1)} - \frac{\partial L_{\mathcal{D}_i}(w_N^i, \phi)}{\partial \phi}\Big|_{(w_2, \phi_2)}\right\|$$

$$\leq \alpha \sum_{m=0}^{N-1} \|Q_m(w_1, \phi_1)U_m(w_1, \phi_1)V_m(w_1, \phi_1) - Q_m(w_2, \phi_2)U_m(w_2, \phi_2)V_m(w_2, \phi_2)\|$$

$$+ \|\nabla_\phi L_{\mathcal{D}_i}(w_N^i(w_1, \phi_1), \phi_1) - \nabla_\phi L_{\mathcal{D}_i}(w_N^i(w_2, \phi_2), \phi_2)\|. \qquad (21)$$

Using the triangle inequality, we have

$$\|Q_m(w_1,\phi_1)U_m(w_1,\phi_1)V_m(w_1,\phi_1) - Q_m(w_2,\phi_2)U_m(w_2,\phi_2)V_m(w_2,\phi_2)\|$$
$$\leq \underbrace{\|Q_m(w_1,\phi_1) - Q_m(w_2,\phi_2)\|\|U_m(w_1,\phi_1)\|\|V_m(w_1,\phi_1)\|}_{R_1}$$
$$+ \underbrace{\|Q_m(w_2,\phi_2)\|\|U_m(w_1,\phi_1) - U_m(w_2,\phi_2)\|\|V_m(w_1,\phi_1)\|}_{R_2}$$
$$+ \underbrace{\|Q_m(w_2,\phi_2)\|\|U_m(w_2,\phi_2)\|\|V_m(w_1,\phi_1) - V_m(w_2,\phi_2)\|}_{R_3}. \tag{22}$$

Combining eq. (21) and eq. (22), we have

$$\left\| \frac{\partial L_{\mathcal{D}_i}(w_N^i,\phi)}{\partial\phi}\Big|_{(w_1,\phi_1)} - \frac{\partial L_{\mathcal{D}_i}(w_N^i,\phi)}{\partial\phi}\Big|_{(w_2,\phi_2)} \right\|$$
$$\leq \alpha \sum_{m=0}^{N-1}(R_1 + R_2 + R_3) + \|\nabla_\phi L_{\mathcal{D}_i}(w_N^i(w_1,\phi_1),\phi_1) - \nabla_\phi L_{\mathcal{D}_i}(w_N^i(w_2,\phi_2),\phi_2)\|. \tag{23}$$

To upper-bound $R_1$, we have

$$R_1 \leq \tau(\|w_m^i(w_1,\phi_1) - w_m^i(w_2,\phi_2)\| + \|\phi_1 - \phi_2\|)(1-\alpha\mu)^{N-m-1}M$$
$$\leq \tau M(1-\alpha\mu)^{N-\frac{m}{2}-1}\|w_1 - w_2\| + \tau M\Big(\frac{2L}{\mu}+1\Big)(1-\alpha\mu)^{N-m-1}\|\phi_1 - \phi_2\|, \tag{24}$$

where the second inequality follows from Lemma 1.

For $R_2$, based on Assumptions 1 and 2, we have

$$R_2 \leq LM\|U_m(w_1,\phi_1) - U_m(w_2,\phi_2)\|. \tag{25}$$

Using the definitions of $U_m(w_1,\phi_1)$ and $U_m(w_2,\phi_2)$ in eq. (19) and using the triangle inequality, we have

$$\|U_m(w_1,\phi_1) - U_m(w_2,\phi_2)\|$$
$$\leq \alpha\|\nabla_w^2 L_{\mathcal{S}_i}(w_{m+1}^i(w_1,\phi_1),\phi_1) - \nabla_w^2 L_{\mathcal{S}_i}(w_{m+1}^i(w_2,\phi_2),\phi_2)\|\|U_{m+1}(w_1,\phi_1)\|$$
$$+ \|I - \alpha\nabla_w^2 L_{\mathcal{S}_i}(w_{m+1}^i(w_1,\phi_1),\phi_1)\|\|U_{m+1}(w_1,\phi_1) - U_{m+1}(w_2,\phi_2)\|$$
$$\leq \alpha\rho(1-\alpha\mu)^{N-m-2}(\|w_{m+1}^i(w_1,\phi_1) - w_{m+1}^i(w_2,\phi_2)\| + \|\phi_1 - \phi_2\|)$$
$$+ (1-\alpha\mu)\|U_{m+1}(w_1,\phi_1) - U_{m+1}(w_2,\phi_2)\|$$
$$\leq \alpha\rho(1-\alpha\mu)^{N-m-2}\Big((1-\alpha\mu)^{\frac{m+1}{2}}\|w_1 - w_2\| + \Big(\frac{2L}{\mu}+1\Big)\|\phi_1 - \phi_2\|\Big)$$
$$+ (1-\alpha\mu)\|U_{m+1}(w_1,\phi_1) - U_{m+1}(w_2,\phi_2)\|,$$

where the last inequality follows from Lemma 1. Telescoping the above inequality over $m$ yields

$$\|U_m(w_1,\phi_1) - U_m(w_2,\phi_2)\|$$
$$\leq (1-\alpha\mu)^{N-m-2}\|U_{N-2}(w_1,\phi_1) - U_{N-2}(w_2,\phi_2)\|$$
$$+ \sum_{t=0}^{N-m-3}(1-\alpha\mu)^t \alpha\rho(1-\alpha\mu)^{N-m-t-2}\Big((1-\alpha\mu)^{\frac{m+t+1}{2}}\|w_1 - w_2\| + \Big(\frac{2L}{\mu}+1\Big)\|\phi_1 - \phi_2\|\Big),$$

which, in conjunction with eq. (19), yields

$$\|U_m(w_1,\phi_1) - U_m(w_2,\phi_2)\| \leq \left(\frac{\alpha\rho}{1-\alpha\mu} + \frac{2\rho}{\mu}\right)(1-\alpha\mu)^{N-1-\frac{m}{2}}\|w_1 - w_2\|$$
$$+ \alpha(N-1-m)\left(\rho + \frac{2\rho L}{\mu}\right)(1-\alpha\mu)^{N-2-m}\|\phi_1 - \phi_2\|. \tag{26}$$

Combining eq. (25) and eq. (26) yields

$$R_2 \leq LM \left( \frac{\alpha\rho}{1-\alpha\mu} + \frac{2\rho}{\mu} \right) (1-\alpha\mu)^{N-1-\frac{m}{2}} \|w_1 - w_2\|$$
$$+ \alpha LM(N-1-m) \left( \rho + \frac{2\rho L}{\mu} \right) (1-\alpha\mu)^{N-2-m} \|\phi_1 - \phi_2\|. \tag{27}$$

For $R_3$, using the triangle inequality, we have

$$R_3 \leq L(1-\alpha\mu)^{N-m-1} L(\|w_N^i(w_1,\phi_1) - w_N^i(w_2,\phi_2)\| + \|\phi_1 - \phi_2\|)$$
$$\leq L^2(1-\alpha\mu)^{\frac{3N}{2}-m-1} \|w_1 - w_2\| + L^2 \left( \frac{2L}{\mu} + 1 \right) (1-\alpha\mu)^{N-1-m} \|\phi_1 - \phi_2\|. \tag{28}$$

where the last inequality follows from Lemma 1.

Combine $R_1, R_2$ and $R_3$ in eq. (24), eq. (27) and eq. (28), we have

$$\sum_{m=0}^{N-1} (R_1 + R_2 + R_3) \leq \frac{2\tau M}{\alpha\mu} (1-\alpha\mu)^{\frac{N-1}{2}} \|w_1 - w_2\| + \frac{\tau M}{\alpha\mu} \left( \frac{2L}{\mu} + 1 \right) \|\phi_1 - \phi_2\|$$
$$+ \frac{2LM}{\alpha\mu} \left( \frac{\alpha\rho}{1-\alpha\mu} + \frac{2\rho}{\mu} \right) (1-\alpha\mu)^{\frac{N-1}{2}} \|w_1 - w_2\| + \frac{\alpha LM}{\alpha^2\mu^2} \left( \rho + \frac{2\rho L}{\mu} \right) \|\phi_1 - \phi_2\|$$
$$+ \frac{L^2}{\alpha\mu} (1-\alpha\mu)^{\frac{N}{2}} \|w_1 - w_2\| + \frac{L^2}{\alpha\mu} \left( \frac{2L}{\mu} + 1 \right) \|\phi_1 - \phi_2\|. \tag{29}$$

In addition, note that

$$\|\nabla_\phi L_{\mathcal{D}_i}(w_N^i(w_1,\phi_1),\phi_1) - \nabla_\phi L_{\mathcal{D}_i}(w_N^i(w_2,\phi_2),\phi_2)\|$$
$$\leq (1-\alpha\mu)^{\frac{N}{2}} L\|w_1 - w_2\| + L \left( \frac{2L}{\mu} + 1 \right) \|\phi_1 - \phi_2\|. \tag{30}$$

Combining eq. (23), eq. (29), and eq. (30) yields

$$\left\| \frac{\partial L_{\mathcal{D}_i}(w_N^i,\phi)}{\partial\phi} \bigg|_{(w_1,\phi_1)} - \frac{\partial L_{\mathcal{D}_i}(w_N^i,\phi)}{\partial\phi} \bigg|_{(w_2,\phi_2)} \right\|$$
$$\leq \left( L + \frac{2\tau M}{\mu} + \frac{2LM}{\mu} \left( \frac{\alpha\rho}{1-\alpha\mu} + \frac{2\rho}{\mu} \right) + \frac{L^2}{\mu} \right) (1-\alpha\mu)^{\frac{N-1}{2}} \|w_1 - w_2\|$$
$$+ \left( L + \frac{\tau M}{\mu} + \frac{LM\rho}{\mu^2} + \frac{L^2}{\mu} \right) \left( \frac{2L}{\mu} + 1 \right) \|\phi_1 - \phi_2\|, \tag{31}$$

which, using an approach similar to eq. (18), completes the proof.

### C.3 Proof of Theorem 1

For notational convenience, we define

$$g_w^i(k) = \frac{\partial L_{\mathcal{D}_i}(w_{k,N}^i,\phi_k)}{\partial w_k}, \quad g_\phi^i(k) = \frac{\partial L_{\mathcal{D}_i}(w_{k,N}^i,\phi_k)}{\partial \phi_k},$$

$$L_w = (1-\alpha\mu)^{\frac{3N}{2}} L + \frac{2\rho M}{\mu} (1-\alpha\mu)^{N-1}, L_w' = \left( L + \alpha\rho MN \right)(1-\alpha\mu)^{N-1} \left( \frac{2L}{\mu} + 1 \right),$$

$$L_\phi = \left( L + \frac{2\tau M}{\mu} + \frac{2LM}{\mu} \left( \frac{\alpha\rho}{1-\alpha\mu} + \frac{2\rho}{\mu} \right) + \frac{L^2}{\mu} \right) (1-\alpha\mu)^{\frac{N-1}{2}},$$

$$L_\phi' = \left( L + \frac{\tau M}{\mu} + \frac{LM\rho}{\mu^2} + \frac{L^2}{\mu} \right) \left( \frac{2L}{\mu} + 1 \right). \tag{32}$$

Then, the updates of Algorithm 1 are given by

$$w_{k+1} = w_k - \frac{\beta_w}{B} \sum_{i\in\mathcal{B}_k} g_w^i(k) \text{ and } \phi_{k+1} = \phi_k - \frac{\beta_\phi}{B} \sum_{i\in\mathcal{B}_k} g_\phi^i(k). \tag{33}$$

Based on the smoothness properties established in eq. (17) and eq. (31) in the proof of Proposition 2, we have

$$L^{meta}(w_{k+1}, \phi_k) \leq L^{meta}(w_k, \phi_k) + \left\langle \frac{\partial L^{meta}(w_k, \phi_k)}{\partial w_k}, w_{k+1} - w_k \right\rangle + \frac{L_w}{2} \|w_{k+1} - w_k\|^2,$$

$$L^{meta}(w_{k+1}, \phi_{k+1}) \leq L^{meta}(w_{k+1}, \phi_k) + \left\langle \frac{\partial L^{meta}(w_{k+1}, \phi_k)}{\partial \phi_k}, \phi_{k+1} - \phi_k \right\rangle + \frac{L'_\phi}{2} \|\phi_{k+1} - \phi_k\|^2.$$

Adding the above two inequalities, we have

$$
\begin{aligned}
L^{meta}(w_{k+1}, \phi_{k+1}) \leq & L^{meta}(w_k, \phi_k) + \left\langle \frac{\partial L^{meta}(w_k, \phi_k)}{\partial w_k}, w_{k+1} - w_k \right\rangle + \frac{L_w}{2} \|w_{k+1} - w_k\|^2 \\
& + \left\langle \frac{\partial L^{meta}(w_k, \phi_k)}{\partial \phi_k}, \phi_{k+1} - \phi_k \right\rangle + \frac{L'_\phi}{2} \|\phi_{k+1} - \phi_k\|^2 \\
& + \left\langle \frac{\partial L^{meta}(w_{k+1}, \phi_k)}{\partial \phi_k} - \frac{\partial L^{meta}(w_k, \phi_k)}{\partial \phi_k}, \phi_{k+1} - \phi_k \right\rangle.
\end{aligned}
\tag{34}
$$

Based on the Cauchy-Schwarz inequality, we have

$$
\begin{aligned}
\left\langle \frac{\partial L^{meta}(w_{k+1}, \phi_k)}{\partial \phi_k} - \frac{\partial L^{meta}(w_k, \phi_k)}{\partial \phi_k}, \phi_{k+1} - \phi_k \right\rangle & \\
\leq & L_\phi \|w_{k+1} - w_k\| \|\phi_{k+1} - \phi_k\| \\
\leq & \frac{L_\phi}{2} \|w_{k+1} - w_k\|^2 + \frac{L_\phi}{2} \|\phi_{k+1} - \phi_k\|^2.
\end{aligned}
\tag{35}
$$

Combining eq. (34) and eq. (35), we have

$$
\begin{aligned}
L^{meta}(w_{k+1}, \phi_{k+1}) \leq & L^{meta}(w_k, \phi_k) + \left\langle \frac{\partial L^{meta}(w_k, \phi_k)}{\partial w_k}, w_{k+1} - w_k \right\rangle + \frac{L_w + L_\phi}{2} \|w_{k+1} - w_k\|^2 \\
& + \left\langle \frac{\partial L^{meta}(w_k, \phi_k)}{\partial \phi_k}, \phi_{k+1} - \phi_k \right\rangle + \frac{L_\phi + L'_\phi}{2} \|\phi_{k+1} - \phi_k\|^2,
\end{aligned}
$$

which, in conjunction with the updates in eq. (33), yields

$$
\begin{aligned}
& L^{meta}(w_{k+1}, \phi_{k+1}) \\
& \leq L^{meta}(w_k, \phi_k) - \left\langle \frac{\partial L^{meta}(w_k, \phi_k)}{\partial w_k}, \frac{\beta_w}{B} \sum_{i \in \mathcal{B}_k} g_w^i(k) \right\rangle + \frac{L_w + L_\phi}{2} \left\| \frac{\beta_w}{B} \sum_{i \in \mathcal{B}_k} g_w^i(k) \right\|^2 \\
& \quad - \left\langle \frac{\partial L^{meta}(w_k, \phi_k)}{\partial \phi_k}, \frac{\beta_\phi}{B} \sum_{i \in \mathcal{B}_k} g_\phi^i(k) \right\rangle + \frac{L_\phi + L'_\phi}{2} \left\| \frac{\beta_\phi}{B} \sum_{i \in \mathcal{B}_k} g_\phi^i(k) \right\|^2.
\end{aligned}
\tag{36}
$$

Let $\mathbb{E}_k = \mathbb{E}(\cdot | w_k, \phi_k)$. Then, conditioning on $w_k, \phi_k$, and taking expectation over eq. (36), we have

$$
\begin{aligned}
\mathbb{E}_k L^{meta}(w_{k+1}, \phi_{k+1}) \overset{(i)}{\leq} & L^{meta}(w_k, \phi_k) - \beta_w \left\| \frac{\partial L^{meta}(w_k, \phi_k)}{\partial w_k} \right\|^2 + \frac{L_w + L_\phi}{2} \mathbb{E}_k \left\| \frac{\beta_w}{B} \sum_{i \in \mathcal{B}_k} g_w^i(k) \right\|^2 \\
& - \beta_\phi \left\| \frac{\partial L^{meta}(w_k, \phi_k)}{\partial \phi_k} \right\| + \frac{L_\phi + L'_\phi}{2} \mathbb{E}_k \left\| \frac{\beta_\phi}{B} \sum_{i \in \mathcal{B}_k} g_\phi^i(k) \right\|^2 \\
\leq & L^{meta}(w_k, \phi_k) - \beta_w \left\| \frac{\partial L^{meta}(w_k, \phi_k)}{\partial w_k} \right\|^2 + \frac{(L_w + L_\phi)\beta_w^2}{2B} \mathbb{E}_k \left\| g_w^i(k) \right\|^2 \\
& + \frac{L_\phi + L_w}{2} \beta_w^2 \left\| \frac{\partial L^{meta}(w_k, \phi_k)}{\partial w_k} \right\|^2 - \beta_\phi \left\| \frac{\partial L^{meta}(w_k, \phi_k)}{\partial \phi_k} \right\|^2 \\
& + \frac{L_\phi + L'_\phi}{2} \left( \frac{\beta_\phi^2}{B} \mathbb{E}_k \left\| g_\phi^i(k) \right\|^2 + \beta_\phi^2 \left\| \frac{\partial L^{meta}(w_k, \phi_k)}{\partial \phi_k} \right\|^2 \right),
\end{aligned}
\tag{37}
$$

where $(i)$ follows from the fact that $\mathbb{E}_k g_w^i(k) = \frac{\partial L^{meta}(w_k, \phi_k)}{\partial w_k}$ and $\mathbb{E}_k g_\phi^i(k) = \frac{\partial L^{meta}(w_k, \phi_k)}{\partial \phi_k}$.

Our next step is to upper-bound $\mathbb{E}_k\big\|g_w^i(k)\big\|^2$ and $\mathbb{E}_k\big\|g_\phi^i(k)\big\|^2$ in eq. (37). Based on the definitions of $g_w^i(k)$ in eq. (32) and using the explicit forms of the meta gradients in Proposition 1, we have

$$
\begin{aligned}
\mathbb{E}_k\big\|g_w^i(k)\big\|^2 &\leq \mathbb{E}_k\bigg\|\prod_{m=0}^{N-1}(I - \alpha\nabla_w^2 L_{\mathcal{S}_i}(w_{k,m}^i, \phi_k))\nabla_w L_{\mathcal{D}_i}(w_{k,N}^i, \phi_k)\bigg\|^2 \\
&\leq (1-\alpha\mu)^{2N}M^2.
\end{aligned}
\tag{38}
$$

Using an approach similar to eq. (38), we have

$$
\begin{aligned}
\mathbb{E}_k\big\|g_\phi^i(k)\big\|^2 &\leq 2\mathbb{E}_k\bigg\|\alpha\sum_{m=0}^{N-1}\nabla_\phi\nabla_w L_{\mathcal{S}_i}(w_{k,m}^i, \phi_k)\prod_{j=m+1}^{N-1}(I - \alpha\nabla_w^2 L_{\mathcal{S}_i}(w_{k,j}^i, \phi_k))\nabla_w L_{\mathcal{D}_i}(w_{k,N}^i, \phi_k)\bigg\|^2 \\
&\quad + 2\|\nabla_\phi L_{\mathcal{D}_i}(w_{k,N}^i, \phi_k)\|^2 \\
&\leq 2\alpha^2 L^2 M^2\mathbb{E}_k\bigg(\sum_{m=0}^{N-1}(1-\alpha\mu)^{N-1-m}\bigg)^2 + 2M^2 \\
&< \frac{2L^2 M^2}{\mu^2} + 2M^2 < 2M^2\left(\frac{L^2}{\mu^2}+1\right).
\end{aligned}
\tag{39}
$$

Substituting eq. (38) and eq. (39) into eq. (37) yields

$$
\begin{aligned}
\mathbb{E}_k L^{meta}(w_{k+1}, \phi_{k+1}) &\leq L^{meta}(w_k, \phi_k) - \left(\beta_w - \frac{L_w + L_\phi}{2}\beta_w^2\right)\left\|\frac{\partial L^{meta}(w_k, \phi_k)}{\partial w_k}\right\|^2 \\
&\quad + \frac{(L_w + L_\phi)\beta_w^2}{2B}(1-\alpha\mu)^{2N}M^2 - \left(\beta_\phi - \frac{L_\phi + L_\phi'}{2}\beta_\phi^2\right)\left\|\frac{\partial L^{meta}(w_k, \phi_k)}{\partial\phi_k}\right\|^2 \\
&\quad + \frac{(L_\phi + L_\phi')\beta_\phi^2}{B}M^2\left(\frac{L^2}{\mu^2}+1\right).
\end{aligned}
\tag{40}
$$

Let $\beta_w = \frac{1}{L_w + L_\phi}$ and $\beta_\phi = \frac{1}{L_\phi + L_\phi'}$. Then, unconditioning on $w_k$ and $\phi_k$ and telescoping eq. (40) over $k$ from $0$ to $K-1$ yield

$$
\begin{aligned}
&\frac{\beta_w}{2}\frac{1}{K}\sum_{k=0}^{K-1}\mathbb{E}\left\|\frac{\partial L^{meta}(w_k, \phi_k)}{\partial w_k}\right\|^2 + \frac{\beta_\phi}{2}\frac{1}{K}\sum_{k=0}^{K-1}\mathbb{E}\left\|\frac{\partial L^{meta}(w_k, \phi_k)}{\partial\phi_k}\right\|^2 \\
&\qquad \leq \frac{L^{meta}(w_0, \phi_0) - \min_{w,\phi} L^{meta}(w, \phi)}{K} + \frac{\beta_w}{2B}(1-\alpha\mu)^{2N}M^2 + \frac{\beta_\phi}{B}M^2\left(\frac{L^2}{\mu^2}+1\right).
\end{aligned}
\tag{41}
$$

Let $\Delta = L^{meta}(w_0, \phi_0) - \min_{w,\phi} L^{meta}(w, \phi)$ and let $\xi$ be chosen from $\{0, ..., K-1\}$ uniformly at random. Then, we have

$$
\begin{aligned}
\mathbb{E}\left\|\frac{\partial L^{meta}(w_\xi, \phi_\xi)}{\partial w_\xi}\right\|^2 &\leq \frac{2\Delta(L_w + L_\phi)}{K} + \frac{(1-\alpha\mu)^{2N}M^2}{B} + \frac{L_w + L_\phi}{L_\phi + L_\phi'}\frac{2}{B}M^2\left(\frac{L^2}{\mu^2}+1\right), \\
\mathbb{E}\left\|\frac{\partial L^{meta}(w_\xi, \phi_\xi)}{\partial\phi_\xi}\right\|^2 &\leq \frac{2\Delta(L_\phi + L_\phi')}{K} + \frac{L_\phi + L_\phi'}{L_w + L_\phi}\frac{1}{B}(1-\alpha\mu)^{2N}M^2 + \frac{2}{B}M^2\left(\frac{L^2}{\mu^2}+1\right),
\end{aligned}
$$

which, in conjunction with the definitions of $L_\phi$, $L_\phi'$ and $L_w$ in eq. (32) and $\alpha = \frac{\mu}{L^2}$, yields

$$
\begin{aligned}
\mathbb{E}\left\|\frac{\partial L^{meta}(w_\xi, \phi_\xi)}{\partial w_\xi}\right\|^2 &\leq \mathcal{O}\left(\frac{\frac{1}{\mu^2}\left(1-\frac{\mu^2}{L^2}\right)^{\frac{N}{2}}}{K} + \frac{\frac{1}{\mu}\left(1-\frac{\mu^2}{L^2}\right)^{\frac{N}{2}}}{B}\right), \\
\mathbb{E}\left\|\frac{\partial L^{meta}(w_\xi, \phi_\xi)}{\partial\phi_\xi}\right\|^2 &\leq \mathcal{O}\left(\frac{\frac{1}{\mu^2}\left(1-\frac{\mu^2}{L^2}\right)^{\frac{N}{2}} + \frac{1}{\mu^3}}{K} + \frac{\frac{1}{\mu}\left(1-\frac{\mu^2}{L^2}\right)^{\frac{3N}{2}} + \frac{1}{\mu^2}}{B}\right).
\end{aligned}
$$

To achieve an $\epsilon$-stationary point, i.e., $\mathbb{E}\left\|\frac{\partial L^{meta}(w,\phi)}{\partial w}\right\|^2 < \epsilon$, $\mathbb{E}\left\|\frac{\partial L^{meta}(w,\phi)}{\partial w}\right\|^2 < \epsilon$, ANIL requires at most

$$KBN = \mathcal{O}\left(\frac{L^2}{\mu^2}\left(1 - \frac{\mu^2}{L^2}\right)^{\frac{N}{2}} + \frac{L^3}{\mu^3}\right)\left(\frac{L}{\mu}\left(1 - \frac{\mu^2}{L^2}\right)^{\frac{3N}{2}} + \frac{L^2}{\mu^2}\right)N\epsilon^{-2}$$

$$\leq \mathcal{O}\left(\frac{N}{\mu^4}\left(1 - \frac{\mu^2}{L^2}\right)^{\frac{N}{2}} + \frac{N}{\mu^5}\right)\epsilon^{-2}$$

gradient evaluations in $w$, $KB = \mathcal{O}\left(\mu^{-4}\left(1 - \frac{\mu^2}{L^2}\right)^{N/2} + \mu^{-5}\right)\epsilon^{-2}$ gradient evaluations in $\phi$, and $KBN = \mathcal{O}\left(\frac{N}{\mu^4}\left(1 - \frac{\mu^2}{L^2}\right)^{N/2} + \frac{N}{\mu^5}\right)\epsilon^{-2}$ evaluations of second-order derivatives.

# D  Proof in Section 3.2: Nonconvex Inner Loop

## D.1  Proof of Proposition 3

Based on the explicit forms of the meta gradient in eq. (10) and using an approach similar to eq. (11), we have

$$\left\|\frac{\partial L_{\mathcal{D}_i}(w_N^i,\phi)}{\partial w}\bigg|_{(w_1,\phi_1)} - \frac{\partial L_{\mathcal{D}_i}(w_N^i,\phi)}{\partial w}\bigg|_{(w_2,\phi_2)}\right\|$$

$$= \left\|\prod_{m=0}^{N-1}(I - \alpha\nabla_w^2 L_{\mathcal{S}_i}(w_m^i(w_1,\phi_1),\phi_1))\nabla_w L_{\mathcal{D}_i}(w_N^i(w_1,\phi_1),\phi_1)\right.$$

$$\left. - \prod_{m=0}^{N-1}(I - \alpha\nabla_w^2 L_{\mathcal{S}_i}(w_m^i(w_2,\phi_2),\phi_2))\nabla_w L_{\mathcal{D}_i}(w_N^i(w_2,\phi_2),\phi_2)\right\|, \qquad (42)$$

where $w_m^i(w,\phi)$ is obtained through the gradient descent steps in eq. (12).

Using the triangle inequality in eq. (42) yields

$$\left\|\frac{\partial L_{\mathcal{D}_i}(w_N^i,\phi)}{\partial w}\bigg|_{(w_1,\phi_1)} - \frac{\partial L_{\mathcal{D}_i}(w_N^i,\phi)}{\partial w}\bigg|_{(w_2,\phi_2)}\right\|$$

$$\leq \left\|\prod_{m=0}^{N-1}(I - \alpha\nabla_w^2 L_{\mathcal{S}_i}(w_m^i(w_2,\phi_2),\phi_2))\right\|\left\|\nabla_w L_{\mathcal{D}_i}(w_N^i(w_1,\phi_1),\phi_1) - \nabla_w L_{\mathcal{D}_i}(w_N^i(w_2,\phi_2),\phi_2)\right\|$$

$$+ \left\|\prod_{m=0}^{N-1}(I - \alpha\nabla_w^2 L_{\mathcal{S}_i}(w_m^i(w_1,\phi_1),\phi_1))\nabla_w L_{\mathcal{D}_i}(w_N^i(w_1,\phi_1),\phi_1)\right.$$

$$\left. - \prod_{m=0}^{N-1}(I - \alpha\nabla_w^2 L_{\mathcal{S}_i}(w_m^i(w_2,\phi_2),\phi_2))\nabla_w L_{\mathcal{D}_i}(w_N^i(w_1,\phi_1),\phi_1)\right\|. \qquad (43)$$

Our next two steps are to upper-bound the two terms at the right hand side of eq. (43), respectively.

Step 1: Upper-bound the first term at the right hand side of eq. (43).

$$\left\|\prod_{m=0}^{N-1}(I - \alpha\nabla_w^2 L_{\mathcal{S}_i}(w_m^i(w_2,\phi_2),\phi_2))\right\|\left\|\nabla_w L_{\mathcal{D}_i}(w_N^i(w_1,\phi_1),\phi_1) - \nabla_w L_{\mathcal{D}_i}(w_N^i(w_2,\phi_2),\phi_2)\right\|$$

$$\overset{(i)}{\leq} (1+\alpha L)^N\left\|\nabla_w L_{\mathcal{D}_i}(w_N^i(w_1,\phi_1),\phi_1) - \nabla_w L_{\mathcal{D}_i}(w_N^i(w_2,\phi_2),\phi_2)\right\|$$

$$\overset{(ii)}{\leq} (1+\alpha L)^N L(\|w_N^i(w_1,\phi_1) - w_N^i(w_2,\phi_2)\| + \|\phi_1 - \phi_2\|), \qquad (44)$$

where $(i)$ follows from the fact that $\|\nabla_w^2 L_{\mathcal{S}_i}(w_m^i(w_2,\phi_2),\phi_2)\| \leq L$, and $(ii)$ follows from Assumption 1. Based on the gradient descent steps in eq. (12), we have, for any $0 \leq m \leq N-1$,

$$w_{m+1}^i(w_1,\phi_1) - w_{m+1}^i(w_2,\phi_2)$$

$$= w_m^i(w_1,\phi_1) - w_m^i(w_2,\phi_2) - \alpha\big(\nabla_w L_{\mathcal{S}_i}(w_m^i(w_1,\phi_1),\phi_1) - \nabla_w L_{\mathcal{S}_i}(w_m^i(w_2,\phi_2),\phi_2)\big).$$

Based on the above equality, we further obtain

$$\|w_{m+1}^i(w_1, \phi_1) - w_{m+1}^i(w_2, \phi_2)\| \leq \|w_m^i(w_1, \phi_1) - w_m^i(w_2, \phi_2)\|$$
$$+ \alpha \|\nabla_w L_{\mathcal{S}_i}(w_m^i(w_1, \phi_1), \phi_1) - \nabla_w L_{\mathcal{S}_i}(w_m^i(w_2, \phi_2), \phi_2)\|$$
$$\leq (1 + \alpha L)\|w_m^i(w_1, \phi_1) - w_m^i(w_2, \phi_2)\| + \alpha L\|\phi_1 - \phi_2\|,$$

where the last inequality follows from Assumption 1. Telescoping the above inequality over $m$ from 0 to $N-1$ yields

$$\|w_N^i(w_1, \phi_1) - w_N^i(w_2, \phi_2)\| \leq (1 + \alpha L)^N \|w_1 - w_2\| + ((1 + \alpha L)^N - 1)\|\phi_1 - \phi_2\|. \quad (45)$$

Combining eq. (44) and eq. (45) yields

$$\left\| \prod_{m=0}^{N-1} (I - \alpha \nabla_w^2 L_{\mathcal{S}_i}(w_m^i(w_2, \phi_2), \phi_2)) \right\| \left\| \nabla_w L_{\mathcal{D}_i}(w_N^i(w_1, \phi_1), \phi_1) - \nabla_w L_{\mathcal{D}_i}(w_N^i(w_2, \phi_2), \phi_2) \right\|$$
$$\leq (1 + \alpha L)^{2N} L(\|w_1 - w_2\| + \|\phi_1 - \phi_2\|). \quad (46)$$

Step 2: Upper-bound the second term at the right hand side of eq. (43).

Based on item 2 in Assumption 1, we have that $\|\nabla_w L_{\mathcal{D}_i}(\cdot, \cdot)\| \leq M$. Then, the second term at the right hand side of eq. (43) is further upper-bounded by

$$M \underbrace{\left\| \prod_{m=0}^{N-1} (I - \alpha \nabla_w^2 L_{\mathcal{S}_i}(w_m^i(w_1, \phi_1), \phi_1)) - \prod_{m=0}^{N-1} (I - \alpha \nabla_w^2 L_{\mathcal{S}_i}(w_m^i(w_2, \phi_2), \phi_2)) \right\|}_{P_{N-1}}. \quad (47)$$

In order to upper-bound $P_{N-1}$ in eq. (47), we define a more general quantity $P_t$ by replacing $N-1$ with $t$ in eq. (47). Based on the triangle inequality, we have

$$P_t \leq \alpha \left\| \prod_{m=0}^{t-1} (I - \alpha \nabla_w^2 L_{\mathcal{S}_i}(w_m^i, \phi_1)) \right\| \left\| \nabla_w^2 L_{\mathcal{S}_i}(w_t^i(w_1, \phi_1), \phi_1) - \nabla_w^2 L_{\mathcal{S}_i}(w_t^i(w_2, \phi_2), \phi_2) \right\|$$
$$+ P_{t-1} \left\| I - \alpha \nabla_w^2 L_{\mathcal{S}_i}(w_t^i(w_2, \phi_2), \phi_2) \right\|$$
$$\leq \alpha(1 + \alpha L)^t \rho(\|w_t^i(w_1, \phi_1) - w_t^i(w_2, \phi_2)\| + \|\phi_1 - \phi_2\|) + (1 + \alpha L)P_{t-1}$$
$$\overset{(i)}{\leq} \alpha \rho(1 + \alpha L)^{2t}(\|w_1 - w_2\| + \|\phi_1 - \phi_2\|) + (1 + \alpha L)P_{t-1},$$

where $(i)$ follows from eq. (45). Rearranging the above inequality, we have

$$P_t - \frac{\rho}{L}(1 + \alpha L)^{2t+1}(\|w_1 - w_2\| + \|\phi_1 - \phi_2\|)$$
$$\leq (1 + \alpha L)(P_{t-1} - \frac{\rho}{L}(1 + \alpha L)^{2t-1}(\|w_1 - w_2\| + \|\phi_1 - \phi_2\|)). \quad (48)$$

Telescoping eq. (48) over $t$ from 1 to $N-1$ yields

$$P_{N-1} - \frac{\rho}{L}(1 + \alpha L)^{2N-1}(\|w_1 - w_2\| + \|\phi_1 - \phi_2\|)$$
$$\leq (1 + \alpha L)^N \left( P_0 - \frac{\rho}{L}(1 + \alpha L)(\|w_1 - w_2\| + \|\phi_1 - \phi_2\|) \right),$$

which, in conjunction with $P_0 = \alpha \|\nabla_w^2 L_{\mathcal{S}_i}(w_1, \phi_1) - \nabla_w^2 L_{\mathcal{S}_i}(w_2, \phi_2)\| \leq \alpha \rho(\|w_1 - w_2\| + \|\phi_1 - \phi_2\|)$, yields

$$P_{N-1} - \frac{\rho}{L}(1 + \alpha L)^{2N-1}(\|w_1 - w_2\| + \|\phi_1 - \phi_2\|)$$
$$\leq (1 + \alpha L)^N \left( \frac{\rho}{L}(\|w_1 - w_2\| + \|\phi_1 - \phi_2\|) \right)$$
$$\leq \frac{\rho}{L}(1 + \alpha L)^{2N-1}(\|w_1 - w_2\| + \|\phi_1 - \phi_2\|), \quad (49)$$

where the last inequality follows because $N \geq 1$. Combining eq. (47), and eq. (49), we have that the second term at the right hand side of eq. (43) is upper-bounded by

$$\frac{2M\rho}{L}(1+\alpha L)^{2N-1}(\|w_1 - w_2\| + \|\phi_1 - \phi_2\|). \tag{50}$$

Step 3: Combine two bounds in Steps 1 and 2.

Combining eq. (46), eq. (50), and using $\alpha < \mathcal{O}(\frac{1}{N})$, we have

$$\left\| \frac{\partial L_{\mathcal{D}_i}(w_N^i, \phi)}{\partial w} \right|_{(w_1, \phi_1)} - \frac{\partial L_{\mathcal{D}_i}(w_N^i, \phi)}{\partial w} \right|_{(w_2, \phi_2)} \right\|$$
$$\leq \left(1 + \alpha L + \frac{2M\rho}{L}\right)(1+\alpha L)^{2N-1}L(\|w_1 - w_2\| + \|\phi_1 - \phi_2\|)$$
$$\leq \text{poly}(M, \rho, \alpha, L)N(\|w_1 - w_2\| + \|\phi_1 - \phi_2\|), \tag{51}$$

which, using an approach similar to eq. (18), completes the proof of the first item in Proposition 3.

We next prove the Lipschitz property of the partial gradient $\frac{\partial L_{\mathcal{D}_i}(w_N^i, \phi)}{\partial \phi}$. Using an approach similar to eq. (21) and eq. (22), we have

$$\left\| \frac{\partial L_{\mathcal{D}_i}(w_N^i, \phi)}{\partial \phi} \right|_{(w_1, \phi_1)} - \frac{\partial L_{\mathcal{D}_i}(w_N^i, \phi)}{\partial \phi} \right|_{(w_2, \phi_2)} \right\|$$
$$\leq \alpha \sum_{m=0}^{N-1}(R_1 + R_2 + R_3) + \|\nabla_\phi L_{\mathcal{D}_i}(w_N^i(w_1, \phi_1), \phi_1) - \nabla_\phi L_{\mathcal{D}_i}(w_N^i(w_2, \phi_2), \phi_2)\|, \tag{52}$$

where $R_1$, $R_2$ and $R_3$ are defined in eq. (22).

To upper-bound $R_1$ in the above inequality, we have

$$R_1 \overset{(i)}{\leq} \tau(\|w_m^i(w_1, \phi_1) - w_m^i(w_2, \phi_2)\| + \|\phi_1 - \phi_2\|)(1+\alpha L)^{N-m-1}M$$
$$\overset{(ii)}{\leq} \tau M(1+\alpha L)^{N-1}(\|w_1 - w_2\| + \|\phi_1 - \phi_2\|), \tag{53}$$

where $(i)$ follows from Assumptions 1 and 2 and $(ii)$ follows from eq. (45).

For $R_2$, using the triangle inequality, we have

$$\|U_m(w_1, \phi_1) - U_m(w_2, \phi_2)\|$$
$$\leq \alpha\|\nabla_w^2 L_{\mathcal{S}_i}(w_{m+1}^i(w_1, \phi_1), \phi_1) - \nabla_w^2 L_{\mathcal{S}_i}(w_{m+1}^i(w_2, \phi_2), \phi_2)\|\|U_{m+1}(w_1, \phi_1)\|$$
$$+ \|I - \alpha\nabla_w^2 L_{\mathcal{S}_i}(w_{m+1}^i(w_1, \phi_1), \phi_1)\|\|U_{m+1}(w_1, \phi_1) - U_{m+1}(w_2, \phi_2)\|$$
$$\leq \alpha\rho(1+\alpha L)^{N-m-2}(\|w_{m+1}^i(w_1, \phi_1) - w_{m+1}^i(w_2, \phi_2)\| + \|\phi_1 - \phi_2\|)$$
$$+ (1+\alpha L)\|U_{m+1}(w_1, \phi_1) - U_{m+1}(w_2, \phi_2)\|$$
$$\leq \alpha\rho(1+\alpha L)^{N-1}(\|w_1 - w_2\| + \|\phi_1 - \phi_2\|)$$
$$+ (1+\alpha L)\|U_{m+1}(w_1, \phi_1) - U_{m+1}(w_2, \phi_2)\|. \tag{54}$$

Telescoping the above inequality over $m$ yields

$$\|U_m(w_1, \phi_1) - U_m(w_2, \phi_2)\| + \frac{\rho}{L}(1+\alpha L)^{N-1}(\|w_1 - w_2\| + \|\phi_1 - \phi_2\|)$$
$$\leq (1+\alpha L)^{N-m-2}\left(\|U_{N-2}(w_1, \phi_1) - U_{N-2}(w_2, \phi_2)\| + \frac{\rho}{L}(1+\alpha L)^{N-1}(\|w_1 - w_2\| + \|\phi_1 - \phi_2\|)\right),$$

which, in conjunction with

$$\|U_{N-2}(w_1, \phi_1) - U_{N-2}(w_2, \phi_2)\| = \alpha\|\nabla_w^2 L_{\mathcal{S}_i}(w_{N-1}^i(w_1, \phi_1), \phi_1) - \nabla_w^2 L_{\mathcal{S}_i}(w_{N-1}^i(w_2, \phi_2), \phi_2)\|$$
$$\leq \alpha\rho(1+\alpha L)^{N-1}(\|w_1 - w_2\| + \|\phi_1 - \phi_2\|),$$

yields that

$$\|U_m(w_1, \phi_1) - U_m(w_2, \phi_2)\| \leq \left(\alpha\rho + \frac{\rho}{L}\right)(1+\alpha L)^{2N-m-3}(\|w_1 - w_2\| + \|\phi_1 - \phi_2\|)$$
$$- \frac{\rho}{L}(1+\alpha L)^{N-1}(\|w_1 - w_2\| + \|\phi_1 - \phi_2\|). \tag{55}$$

Based on Assumption 1, we have $\|Q_m(w_2, \phi_2)\| \leq L$ and $\|V_m(w_1, \phi_1)\| \leq M$, which, combined with eq. (55) and the definition of $R_2$ in eq. (22), yields

$$
\begin{aligned}
R_2 \leq & ML\Big(\alpha\rho + \frac{\rho}{L}\Big)(1 + \alpha L)^{2N-m-3}(\|w_1 - w_2\| + \|\phi_1 - \phi_2\|) \\
& - M\rho(1 + \alpha L)^{N-1}(\|w_1 - w_2\| + \|\phi_1 - \phi_2\|).
\end{aligned}
\tag{56}
$$

For $R_3$, using Assumption 1, we have

$$
\begin{aligned}
R_3 \leq & L(1 + \alpha L)^{N-m-1}\|\nabla_w L_{\mathcal{D}_i}(w_N^i(w_1, \phi_1), \phi_1) - \nabla_w L_{\mathcal{D}_i}(w_N^i(w_2, \phi_2), \phi_2)\| \\
\leq & L^2(1 + \alpha L)^{2N-m-1}(\|w_1 - w_2\| + \|\phi_1 - \phi_2\|),
\end{aligned}
\tag{57}
$$

where the last inequality follows from eq. (45). Combining eq. (53), eq. (56) and eq. (57) yields

$$
\begin{aligned}
R_1 + R_2 + R_3 \leq & M(\tau - \rho)(1 + \alpha L)^{N-1}(\|w_1 - w_2\| + \|\phi_1 - \phi_2\|) \\
& + M\rho(1 + \alpha L)^{2N-m-2}(\|w_1 - w_2\| + \|\phi_1 - \phi_2\|) \\
& + L^2(1 + \alpha L)^{2N-m-1}(\|w_1 - w_2\| + \|\phi_1 - \phi_2\|).
\end{aligned}
\tag{58}
$$

Combining eq. (52), eq. (58), and using eq. (45) and $\alpha < \mathcal{O}(\frac{1}{N})$, we have

$$
\begin{aligned}
& \left\| \frac{\partial L_{\mathcal{D}_i}(w_N^i, \phi)}{\partial \phi}\Big|_{(w_1, \phi_1)} - \frac{\partial L_{\mathcal{D}_i}(w_N^i, \phi)}{\partial \phi}\Big|_{(w_2, \phi_2)} \right\| \\
& \leq \Big(\alpha M(\tau - \rho)N(1 + \alpha L)^{N-1} + \Big(L + \frac{\rho M}{L}\Big)(1 + \alpha L)^{2N}\Big)(\|w_1 - w_2\| + \|\phi_1 - \phi_2\|) \\
& \leq \text{poly}(M, \rho, \tau, \alpha, L)N(\|w_1 - w_2\| + \|\phi_1 - \phi_2\|),
\end{aligned}
\tag{59}
$$

which, using an approach similar to eq. (18), finishes the proof of the second item in Proposition 3.

## D.2 Proof of Theorem 2

For notational convenience, we define

$$
\begin{aligned}
g_w^i(k) &= \frac{\partial L_{\mathcal{D}_i}(w_{k,N}^i, \phi_k)}{\partial w_k}, \quad g_\phi^i(k) = \frac{\partial L_{\mathcal{D}_i}(w_{k,N}^i, \phi_k)}{\partial \phi_k}, \\
L_w &= \big(L + \alpha L^2 + 2M\rho\big)(1 + \alpha L)^{2N-1}, \\
L_\phi &= \alpha M(\tau - \rho)N(1 + \alpha L)^{N-1} + \Big(L + \frac{\rho M}{L}\Big)(1 + \alpha L)^{2N}.
\end{aligned}
\tag{60}
$$

Based on the smoothness properties established in eq. (51) and eq. (59) in the proof of Proposition 3, we have

$$
\begin{aligned}
L^{meta}(w_{k+1}, \phi_k) &\leq L^{meta}(w_k, \phi_k) + \Big\langle \frac{\partial L^{meta}(w_k, \phi_k)}{\partial w_k}, w_{k+1} - w_k \Big\rangle + \frac{L_w}{2}\|w_{k+1} - w_k\|^2, \\
L^{meta}(w_{k+1}, \phi_{k+1}) &\leq L^{meta}(w_{k+1}, \phi_k) + \Big\langle \frac{\partial L^{meta}(w_{k+1}, \phi_k)}{\partial \phi_k}, \phi_{k+1} - \phi_k \Big\rangle + \frac{L_\phi}{2}\|\phi_{k+1} - \phi_k\|^2.
\end{aligned}
$$

Adding the above two inequalities, and using an approach similar to eq. (36), we have

$$
\begin{aligned}
& L^{meta}(w_{k+1}, \phi_{k+1}) \\
& \leq L^{meta}(w_k, \phi_k) - \Big\langle \frac{\partial L^{meta}(w_k, \phi_k)}{\partial w_k}, \frac{\beta_w}{B}\sum_{i \in \mathcal{B}_k} g_w^i(k) \Big\rangle + \frac{L_w + L_\phi}{2}\Big\| \frac{\beta_w}{B}\sum_{i \in \mathcal{B}_k} g_w^i(k) \Big\|^2 \\
& \quad - \Big\langle \frac{\partial L^{meta}(w_k, \phi_k)}{\partial \phi_k}, \frac{\beta_\phi}{B}\sum_{i \in \mathcal{B}_k} g_\phi^i(k) \Big\rangle + L_\phi\Big\| \frac{\beta_\phi}{B}\sum_{i \in \mathcal{B}_k} g_\phi^i(k) \Big\|^2.
\end{aligned}
\tag{61}
$$

Let $\mathbb{E}_k = \mathbb{E}(\cdot | w_k, \phi_k)$. Then, conditioning on $w_k, \phi_k$, taking expectation over eq. (61) and using an approach similar to eq. (37), we have

$$
\begin{aligned}
\mathbb{E}_k L^{meta}(w_{k+1}, \phi_{k+1}) \leq & L^{meta}(w_k, \phi_k) - \beta_w \left\| \frac{\partial L^{meta}(w_k, \phi_k)}{\partial w_k} \right\|^2 + \frac{(L_w + L_\phi)\beta_w^2}{2B} \mathbb{E}_k \left\| g_w^i(k) \right\|^2 \\
& + \frac{L_\phi + L_w}{2} \beta_w^2 \left\| \frac{\partial L^{meta}(w_k, \phi_k)}{\partial w_k} \right\|^2 - \beta_\phi \left\| \frac{\partial L^{meta}(w_k, \phi_k)}{\partial \phi_k} \right\|^2 \\
& + L_\phi \left( \frac{\beta_\phi^2}{B} \mathbb{E}_k \left\| g_\phi^i(k) \right\|^2 + \beta_\phi^2 \left\| \frac{\partial L^{meta}(w_k, \phi_k)}{\partial \phi_k} \right\|^2 \right).
\end{aligned}
\tag{62}
$$

Our next step is to upper-bound $\mathbb{E}_k \left\| g_w^i(k) \right\|^2$ and $\mathbb{E}_k \left\| g_\phi^i(k) \right\|^2$ in eq. (62). Based on the definitions of $g_w^i(k)$ in eq. (60) and Proposition 1, we have

$$
\begin{aligned}
\mathbb{E}_k \left\| g_w^i(k) \right\|^2 \leq & \mathbb{E}_k \left\| \frac{\partial L_{\mathcal{D}_i}(w_{k,N}^i, \phi_k)}{\partial w_k} \right\|^2 = \mathbb{E}_k \left\| \prod_{m=0}^{N-1} (I - \alpha \nabla_w^2 L_{\mathcal{S}_i}(w_{k,m}^i, \phi_k)) \nabla_w L_{\mathcal{D}_i}(w_{k,N}^i, \phi_k) \right\|^2 \\
\leq & \mathbb{E}_k (1 + \alpha L)^{2N} M^2 = (1 + \alpha L)^{2N} M^2.
\end{aligned}
\tag{63}
$$

Using an approach similar to eq. (63), we have

$$
\begin{aligned}
\mathbb{E}_k \left\| g_\phi^i(k) \right\|^2 \leq & 2 \mathbb{E}_k \left\| \alpha \sum_{m=0}^{N-1} \nabla_\phi \nabla_w L_{\mathcal{S}_i}(w_{k,m}^i, \phi_k) \prod_{j=m+1}^{N-1} (I - \alpha \nabla_w^2 L_{\mathcal{S}_i}(w_{k,j}^i, \phi_k)) \nabla_w L_{\mathcal{D}_i}(w_{k,N}^i, \phi_k) \right\|^2 \\
& + 2 \| \nabla_\phi L_{\mathcal{D}_i}(w_{k,N}^i, \phi_k) \|^2 \\
\leq & 2 \alpha^2 L^2 M^2 \mathbb{E}_k \left( \sum_{m=0}^{N-1} (1 + \alpha L)^{N-1-m} \right)^2 + 2M^2 \\
< & 2M^2 (1 + \alpha L)^N - 1)^2 + 2M^2 < 2M^2 (1 + \alpha L)^{2N}.
\end{aligned}
\tag{64}
$$

Substituting eq. (63) and eq. (64) into eq. (62), we have

$$
\begin{aligned}
\mathbb{E}_k L^{meta}(w_{k+1}, \phi_{k+1}) \leq & L^{meta}(w_k, \phi_k) - \left( \beta_w - \frac{L_w + L_\phi}{2} \beta_w^2 \right) \left\| \frac{\partial L^{meta}(w_k, \phi_k)}{\partial w_k} \right\|^2 \\
& + \frac{(L_w + L_\phi)\beta_w^2}{2B} (1 + \alpha L)^{2N} M^2 - \left( \beta_\phi - L_\phi \beta_\phi^2 \right) \left\| \frac{\partial L^{meta}(w_k, \phi_k)}{\partial \phi_k} \right\|^2 \\
& + \frac{2 L_\phi \beta_\phi^2}{B} (1 + \alpha L)^{2N} M^2.
\end{aligned}
\tag{65}
$$

Set $\beta_w = \frac{1}{L_w + L_\phi}$ and $\beta_\phi = \frac{1}{2 L_\phi}$. Then, unconditioning on $w_k, \phi_k$ in eq. (65), we have

$$
\begin{aligned}
\mathbb{E} L^{meta}(w_{k+1}, \phi_{k+1}) \leq & \mathbb{E} L^{meta}(w_k, \phi_k) - \frac{\beta_w}{2} \mathbb{E} \left\| \frac{\partial L^{meta}(w_k, \phi_k)}{\partial w_k} \right\|^2 + \frac{\beta_w}{2B} (1 + \alpha L)^{2N} M^2 \\
& - \frac{\beta_\phi}{2} \mathbb{E} \left\| \frac{\partial L^{meta}(w_k, \phi_k)}{\partial \phi_k} \right\|^2 + \frac{\beta_\phi}{B} (1 + \alpha L)^{2N} M^2.
\end{aligned}
$$

Telescoping the above equality over $k$ from 0 to $K - 1$ yields

$$
\begin{aligned}
& \frac{\beta_w}{2} \frac{1}{K} \sum_{k=0}^{K-1} \mathbb{E} \left\| \frac{\partial L^{meta}(w_k, \phi_k)}{\partial w_k} \right\|^2 + \frac{\beta_\phi}{2} \frac{1}{K} \sum_{k=0}^{K-1} \mathbb{E} \left\| \frac{\partial L^{meta}(w_k, \phi_k)}{\partial \phi_k} \right\|^2 \\
& \qquad \leq \frac{L^{meta}(w_0, \phi_0) - \min_{w, \phi} L^{meta}(w, \phi)}{K} + \frac{\beta_w + 2\beta_\phi}{2B} (1 + \alpha L)^{2N} M^2.
\end{aligned}
\tag{66}
$$

Let $\Delta = L^{meta}(w_0, \phi_0) - \min_{w,\phi} L^{meta}(w, \phi) > 0$ and let $\xi$ be chosen from $\{0, ..., K-1\}$ uniformly at random. Then, eq. (66) further yields

$$\mathbb{E}\left\|\frac{\partial L^{meta}(w_\xi, \phi_\xi)}{\partial w_\xi}\right\|^2 \leq \frac{2\Delta(L_w + L_\phi)}{K} + \frac{1 + \frac{L_w + L_\phi}{L_\phi}}{B}(1 + \alpha L)^{2N}M^2$$

$$\mathbb{E}\left\|\frac{\partial L^{meta}(w_\xi, \phi_\xi)}{\partial \phi_\xi}\right\|^2 \leq \frac{4\Delta L_\phi}{K} + \frac{2 + \frac{2L_\phi}{L_w + L_\phi}}{B}(1 + \alpha L)^{2N}M^2,$$

which, in conjunction with the definitions of $L_w$ and $L_\phi$ in eq. (60) and using $\alpha < \mathcal{O}(\frac{1}{N})$, yields

$$\mathbb{E}\left\|\frac{\partial L^{meta}(w_\xi, \phi_\xi)}{\partial w_\xi}\right\|^2 \leq \mathcal{O}\left(\frac{N}{K} + \frac{N}{B}\right),$$

$$\mathbb{E}\left\|\frac{\partial L^{meta}(w_\xi, \phi_\xi)}{\partial \phi_\xi}\right\|^2 \leq \mathcal{O}\left(\frac{N}{K} + \frac{N}{B}\right). \tag{67}$$

To achieve an $\epsilon$-stationary point, i.e., $\mathbb{E}\left\|\frac{\partial L^{meta}(w,\phi)}{\partial w}\right\|^2 < \epsilon, \mathbb{E}\left\|\frac{\partial L^{meta}(w,\phi)}{\partial w}\right\|^2 < \epsilon$, $K$ and $B$ need to be at most $\mathcal{O}(N\epsilon^{-2})$, which, in conjunction with the gradient forms in Proposition 1, completes the complexity results.