[Reviews · NeurIPS 2020]

Review 1

Summary and Contributions: The paper proposes a convergence analysis for Almost No Inner-Loop (ANIL), a type of MAML. The analysis shows ANIL convergence in both strongly- and non-convexity. Moreover, computational complexity of ANIL w.r.t. to the number of inner-step is also discussed. The experiments validate the proposed theoretical foundation. The conclusion of this analysis is that ANIL provides a more efficient approach compared to MAML.

Strengths: - The convergence of ANIL is discussed under general assumptions: strongly and non-convex. - The comparison between ANIL and MAML is clear. - The experiments confirm the proposed theoretical analysis. - The paper has clear focuses and complexity comparison to the meta-learning algorithms. - The finding between non-convex and strong-convex geometries regarding the number of steps is very important when training the meta-learning algorithm and updating only the classifier.

Weaknesses: - It is not very clear about the evaluation for outer iterations. Is the number of aggregated tasks affecting the convergence too? In MAML, the gradient for outer-loop is computed based on the inner-loop of several tasks. - An experiment (the same setup as in Fig. 1 and 2) about the convergence comparison between ANIL and MAML will strengthen the analysis. - Another experiment w.r.t. the number of sample $B$ will be insightful. Particularly, the number of samples in the support set and the query set (on mini-ImageNet) may affect the error and the convergence. - I would like to know the reason why the number of samples in the inner-loop is not taken into account for convergence analysis? Fallah et al. [4] considers this number of samples in the inner-loop to their analysis. - The inner-loop stepsize is different between non-convex and strongly convex cases. Non-convex stepsize is always larger than strongly-convex stepsize. Do the authors have experiments with the same step size? Because, large stepsize in the non-convex setting may cause suboptimal performance if the inner-loop has many iterations. - Line 170: "From an optimization perspective, this allows a larger step size chosen for the outer-loop meta optimization, and hence yields a faster convergence rate". Could the authors elaborate more to this sentence? Because large stepsize does not always correlate directly to faster convergence, especially, in the case of few samples (facing the noisy gradients).

Correctness: The analysis is correct. The experiments follow the common setup to evaluate the classification performance for meta-learning.

Clarity: The paper has clear purposes and focuses. The comparison is also clear in that there is comparison to the meta-learning approach called MAML in Table 2.

Relation to Prior Work: ANIL and MAML are mentioned with an introduction. The algorithms for both of them are correct and adequately discussed in the paper, so the paper is self-contained.

Reproducibility: Yes

Additional Feedback: After reading the rebuttal and other reviews, I am inclined to accept this work. The analysis of convergence related to the number of step is indeed important to the optimization based meta-learning. The rebuttal also addressed my concerns.


Review 2

Summary and Contributions: --==Update==-- I thank the authors for the feedback. I appreciate that you walked me through the places where the extra conditioning comes from, but my question was rather if it can be improved. For instance, a simple theoretical example, such as a 1-dimensional quadratic objective, could elaborate on the tightness. Alternatively, we could see tightness from a simple empirical example. Nothing of this type was provided by the authors. Thus, I believe my concerns were not properly addressed and keep my score as is. ----------- This work presents an analysis of ANIL--a recently proposed method for model-agnostic meta-learning that does not require updating all parameters inside the inner loop. The theory considers a general inner loop of length N and gives convergence guarantees for strongly convex and nonconvex objectives. The only serious issue that I see in this work is the tightness of the results: they depend on the conditioning kappa as kappa^2 inside the contraction in Theorem 2 and as kappa^3 in Proposition 2. I

Strengths: Meta-learning has gained a lot of attentions due to its effectiveness in practice and the work is likely to be of interest to the community. The work is easy to follow and the obtained theory is meaningful. The theoretical findings are supported by a couple of numerical experiments.

Weaknesses: I have strong doubts about optimality of the analysis. The dependence on the conditioning is pretty bad even for a non-accelerated method. Moreover, iMAML has sqrt(kappa) dependence in contrast to the poly(kappa) bounds of this work. The assumptions require smoothness of the objective and Lipschitz Hessians, which is rather standard, but the Lipschitzness of the objective seems to be an extra assumption compared to iMAML. The proofs are also quite complicated, which may limit their future extensions. However, this is a small issue as the results are of interest on their own. The experiments did not try to validate the theory and only showed how the choice of N affects convergence for strongly convex and nonconvex training. In both cases, a neural network was used to run the experiments. I think it would be nice to have an experiments where the authors control more parameters and can see, for example, if the dependence on the conditioning proposed in the theory is tight. This can be done even with a toy problem, not necessarily a neural network.

Correctness: I'm pretty confident about correctness of Proposition 1, but it was pretty difficult to verify Propositions 2 and 3. However, I couldn't find any mistake and the steps seem logical and to some extent follow the analysis of standard stochastic gradient descent.

Clarity: The work is very well written. I enjoyed reading the main section of the paper, although the proofs are quite difficult to follow. I'm not sure if it's an issue of bad notation (e.g., L^{meta} takes a lot of space) or it is really required, but many equations are extremely long. I also think the proofs should be simplified. For example, the proof of proposition 3 takes more than 3 pages and consists of multiple steps. This can be decomposed into a couple of lemmas that can be also explained more properly.

Relation to Prior Work: This submission has a great overview of related work and covers all recent works on the topic that I know. Since this work does not propose a new method and instead provides an analysis of an existing method (i.e., ANIL), I think it is fine that the authors did not do numerical comparison with other meta-learning methods. Two small issues. 1) reference [13] is outdated: the work has changed its title and has been published at ICML. 2) I think works [24] and [31] should be also added to the tables with complexities for completeness, especially since [31] proposed a similar multi-step method (even though it updates all parameters).

Reproducibility: Yes

Additional Feedback: The role of N as presented in the theory seems to make convergence only slower. As the authors note, the complexity has exponential and linear dependence on N, so N should be sufficiently small. However, it appears from the bound that it only makes sense to use N=2 because 1) the linear term even has larger dependence on mu (mu^5 instead of mu^4) and 2) the term inside exponent is 1-mu^2/L^2 which is extremely close to 1. At the same time, optimality of N=1 contradicts the experiments that show values N=4 and N=7 to perform the best. Can the authors clarifty on this aspect? I feel like the authors somewhat abused the "Broader Impact" section to get over 8 pages and write their conclusion there. I think this is not what it's meant to be and the authors should address in this section the potential issues of using meta learning and ANIL in particular, if they see any.


Review 3

Summary and Contributions: As an efficient version of MAML, ANIL was proposed recently by Raghu et al. 2019, but the theoretical convergence of ANIL has not been studied yet. This work provide theoretical convergence guarantee for the ANIL algorithm under strongly-convex and nonconvex inner-loop loss functions, respectively. ------------ UPDATE ----------------- I have read the author's response. All my concerns have been addressed and I think this paper should be accepted.

Strengths: This work reveals different performance behaviors of ANIL under the two geometries by characterizing the impact of inner-loop adaptation steps on the overall convergence rate. 1. soundness of the claims a. theoretical grounding: this work characterize the convergence rate and the computational complexity for ANIL with N-step inner-loop gradient descent, under nonconvex outer-loop loss geometry, and under two representative inner-loop loss geometries, i.e., strongly-convexity and nonconvexity. Technical parts sounds good to me. b. empirical evaluation: theory were validated over two benchmarks for few-shot multiclass classification, FC100 and miniImageNet. The results are solid. 2. significance and novelty of the contribution The result could further provide guidelines for the hyper-parameter selections for ANIL under different inner-loop loss geometries. This type of results are significant and novel. 3. relevance to the NeurIPS community The work is higly related to the NeurIPS and ML community.

Weaknesses: Generally, I feel comfortable with this paper. It would be even better to give some key proof steps in the paper. But consider the page limit, it is also ok to keep the current form.

Correctness: The claims and methods sound correct to me. The empirical methodology and results sound correct, too.

Clarity: the paper is well written and easy to read

Relation to Prior Work: it is clealy discussed how this work differs from previous contributions

Reproducibility: Yes

Additional Feedback:


Review 4

Summary and Contributions: This paper proves convergence rate and computational complexity of ANIL where only part of the parameters are being updated during MAML and shows that empirically the convergence of ANIL follows what the theory predicts. The paper also shows the geometry of the inner-loop has a significant impact on the convergence of ANIL. ------------------------------------Update-------------------------------------- Most of my concerns have been addressed in authors' feedback and I think this paper should be accepted.

Strengths: Intuitively, ANIL should be much more efficient than MAML but despite the empirical evidence there was no theoretical analysis of the convergence rate. This paper closes the gap between intuition and theory, and shows that ANIL is indeed more efficient when it comes to convergence and computational efficiency. It also highlights the difference between non-convex and convex inner-loop structure and how it affects convergence. This is important because updating partial parameters is a standard practice when MAML is being applied to models with many parameters such as convolutional neural networks. The analysis to my knowledge is novel and this paper is of interest to the optimization community and represents a step toward better understanding various trade-off and properties of gradient-based meta learning.

Weaknesses: One weakness of this paper is that it does not offer any insight into improving the existing methods other than providing some guidelines on choosing the number of inner-loop steps. While the exact theoretical guarantee is indeed, these conclusions can be reached almost without looking at the exact theory. Another major issue I have is that the convergence analysis is done with vanilla gradient descent but all experiments are done with Adam which is an adaptive method. This makes the experiments much less compelling since adaptive methods significantly change the convergence behavior of optimization problems. It would be good to see experiments with purely first order methods even if they are on simpler tasks such as sinusoidal regression.

Correctness: The claims and derivation look reasonable but I did not thoroughly check the proof. Empirical results could be improved. Specifically, Since the performance difference is relatively small, it would be nice to see the experimental results over different random seeds. Furthermore, it would also be nice to see the results over different hyperparameter settings (e.g. architecture or without batch normalization) since they also significantly affect the optimization landscape. There is also discrepancy between the theory and empirical results mentioned above which should be addressed.

Clarity: The paper is well-written and easy to follow.

Relation to Prior Work: Yes.

Reproducibility: Yes

Additional Feedback:

[Author Response · NeurIPS 2020]

**Reviewer 1:** **Unclear about the evaluation for outer iterations; Does the number of aggregated tasks affect**
**convergence:** *Great question! Yes, the total complexity is proportional to the number of aggregated tasks. In addition,*
*in terms of updating task-specific parameters, ANIL takes the same steps as MAML, and the outer-loop gradient (line*
10 *of Alg.* 1*) also depends on the* inner-loop *outputs* $w_{k,N}^i$ *of tasks in* $\mathcal{B}_k$*. We will clarify it in the revision.*

**Add experiments to compare ANIL and MAML and w.r.t. the size** $B$ **of samples:** *Thanks for the suggestion! We*
*will absolutely follow these suggestions to add experiments in the revision.*

**Why sample size in inner-loop is not taken into analysis, as Fallah et al. [4] does:** *Great question! In our setting,*
*the inner-loop loss functions take a* finite-sum *form over* pre-assigned *samples. As a result, the inner-loop updates take*
*full gradient descent without data sampling, and hence gradient estimation bias (which can introduce sample size) does*
not *exist in convergence bound. This setting has also been considered in Rajeswaran et al. [24], Ji et al. [13]. As a*
*comparison, Fallah et al. [4] considered a different setting, where loss functions take the form* in expectation *and fresh*
*data are sampled as the algorithm runs. As a result, their analysis involves an estimation bias, which introduces the*
*dependence on the number of samples.*

**Experiments for non-convex and strongly convex cases with the same stepsize:** *Great point! We have run more*
*experiments on FC100 with the same stepsize* 0.03, 0.05, 0.1 *for both cases, and the nature of results remain the same.*

**Elaborate more for line 170:** *The statement specifically refers to Theorem 1, where increasing* $N$ *leads to larger*
*stepsize* $\beta_w$*, which yields faster convergence rate* $\mathcal{O}(\frac{1}{K\beta_w})$*. We will clarify it in the revision.*

**Reviewer 2:** **Dependence on** $\kappa$**. iMAML depends on** $\sqrt{\kappa}$ **in contrast to poly**$(\kappa)$ **of this work:** *Great question!*
*High-level speaking, better dependence on* $\kappa$ *for iMAML is based on an ideal solution of an inner-loop optimization*
*problem, which can take many iterations. ANIL takes only a few inner-loop iterations (thus a lower cost), but has*
*worse outer-loop convergence (in terms of* $\kappa$*). Technically speaking, smoothness analysis of iMAML upper-bounds*
*the distance between* two optimal points $w_*^i(w_1)$ *and* $w_*^i(w_2)$*, each obtained by solving an inner-loop optimization*
*problem. As a comparison, analysis of ANIL upper-bounds the distance between* two inner-loop paths*, which sums up*
*the distances between* all *corresponding points on the two paths (see eq.* (21)*). This results in a worse dependence in* $\kappa$*.*

**Add an experiment to verify the tightness:** *Great point! We will definitely add such an experiment in the revision.*

**Extra assumption on Lipschitzness of the objective, which is not for iMAML:** *We take this assumption to ensure*
*the meta gradient to be bounded. As a comparison, iMAML alternatively assumes the search space of parameters to be*
*bounded (see Theorem 1 therein) so that the meta gradient (eq. (5) therein) can be bounded.*

**The role of** $N$ **in the theory seems to make convergence only slower:** *The exponential term has a worse dependence*
*on constants and* $\tau, M$ *than the linear term (we will add explicit forms in the revision), and hence the choice of* $N$
*depends on how large* $\kappa$ *is. For large* $\kappa$*, as the reviewer also pointed out, a small* $N = 2$ *is a better choice. However,*
*when* $\kappa$ *is not very large, e.g., in our experiments (in which increasing* $N$ *accelerates the iteration rate), the exponential*
*term dominates for a small* $N$*, and hence a larger* $N$ *is preferred. We will clarify it in the revision.*

**Optimality of** $N = 1$ **contradicts the experiments where** $N = 4, 7$ **are the best:** *We assume the reviewer refers*
*to the experiments in left plot of Figure 2(a). This can be due to the fact that the influence of* $N$ *w.r.t.* the number
of outer-loop iterations *is offset by other constant-level parameters for small* $N$*. Evidently, right plot of Figure 2(a)*
*indicates that* $N = 1$ *is optimal w.r.t.* the running time*, which agrees with our result on computational complexity.*

**Suggestions on presentation and references:** *Many thanks! We will follow these suggestions to improve our paper.*

**Reviewer 3:** *We thank the reviewer for the positive comments!*

**Reviewer 4:** **Comments on insight of theoretical results:** *Our results theoretically characterize the order-level*
*computational complexity for ANIL and its comparison to MAML. In addition, our analysis techniques can be useful for*
*developing guarantee for other meta-learning and more broadly bi-level optimization algorithms.*

**Convergence analysis is done with vanilla gradient descent but**
**all experiments are done with Adam; Experiments with purely**
**first-order methods:** *Great point! We have done new experiments*
*on FC100 dataset using* mini-batch SGD *with a learning rate of*
0.05*, and the results are given in the figures to the right. It can be*
*seen that the nature of the results remains the same as those in our*
*paper. More results will be added in the revision.*

**Run experiments over different random seeds and over different hyper-parameter settings:** *Many thanks! We*
*will definitely provide these experimental results in the revision.*

[Meta-Review · NeurIPS 2020]

This paper studies the convergence rate and computational complexity of ANIL (a variant of MAML) for cases of strongly-convex and nonconvex inner-loop loss. The paper focuses on an important problem (due to increasing interest in MAML type methods) and it empirically backups its theoretical claims. There were some concerns initially, specially those raised by R4 (providing no insight into improving the existing methods, discrepancy between optimization methods in theoretical analysis and empirical verification). However, authors' response was very helpful and at the end all reviewers agree that the submission is ready for publication. I strongly recommend authors' to incorporate R1's post rebuttal comment in the final version of this work, as it can be an important and yet easy to add component. I am referring to R1's request: " a simple theoretical example, such as a 1-dimensional quadratic objective, could elaborate on the tightness. Alternatively, we could see tightness from a simple empirical example. Nothing of this type was provided by the authors".